# Re-Evaluating the Impact of Unseen-Class Unlabeled Data on Semi-Supervised Learning Model

**Rundong He**[2*] **Yicong Dong**[2] **Lanzhe Guo**[3] **Yilong Yin**[2†] **Tailin Wu**[1†]

[1] Department of Artificial Intelligence, School of Engineering, Westlake University
[2] School of Software, Shandong University
[3] School of Intelligence Science and Technology, Nanjing University
`rundong_he@mail.sdu.edu.cn, ylyin@sdu.edu.cn, wutailin@westlake.edu.cn`

## Abstract

Semi-supervised learning (SSL) effectively leverages unlabeled data and has been proven successful across various fields. Current safe SSL methods believe that unseen classes in unlabeled data harm the performance of SSL models. However, previous methods for assessing the impact of unseen classes on SSL model performance are flawed. They fix the size of the unlabeled dataset and adjust the proportion of unseen classes within the unlabeled data to assess the impact. This process contravenes the principle of controlling variables. Adjusting the proportion of unseen classes in unlabeled data alters the proportion of seen classes, meaning the decreased classification performance of seen classes may not be due to an increase in unseen class samples in the unlabeled data, but rather a decrease in seen class samples. Thus, the prior flawed assessment standard that "unseen classes in unlabeled data can damage SSL model performance" may not always hold true. This paper strictly adheres to the principle of controlling variables, maintaining the proportion of seen classes in unlabeled data while only changing the unseen classes across five critical dimensions, to investigate their impact on SSL models from global robustness and local robustness. Experiments demonstrate that unseen classes in unlabeled data do not necessarily impair the performance of SSL models; in fact, under certain conditions, unseen classes may even enhance them.

## 1 Introduction

Semi-supervised learning (SSL) has achieved tremendous success across various fields. For instance, in the realm of natural language processing, SSL has been instrumental in improving language models. An example is the development of advanced text classifiers that can understand context and sentiment with minimal labeled data. Another field where it has made a significant impact is image recognition. Here, SSL techniques have enabled the creation of more accurate and efficient image classification models, which can recognize and categorize images with limited human-annotated examples. These successes demonstrate the effectiveness of SSL in handling complex tasks in different domains.

SSL effectively leverages unlabeled data and has been proven successful across various fields. Current safe SSL methods generally believe that unseen classes in unlabeled data will damage the performance of SSL models. Therefore, their common practice is to identify and either remove these unseen classes from the unlabeled data or assign them lower weights. Guo *et al.* Guo et al. (2020) assigned lower weights to possible unseen-class unlabeled data by bi-level optimization. OpenMatch Saito et al. (2021) combines FixMatch Sohn et al. (2020) and a one-vs-all classifier, enhancing the model's anomaly detection capabilities by introducing an open set soft consistency regularization loss. Fix_A_Step Huang et al. (2023a) adjusts the relationship between labeled and unlabeled data from a gradient perspective by integrating MixUp Zhang et al. (2017) technology with an update direction correction strategy. However, previous safe SSL methods for assessing the impact of unseen

---

*Work done as an intern at Westlake University
†Corresponding authors

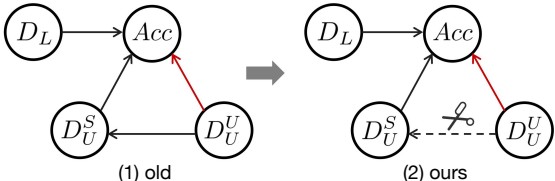

Figure 1: The structural causal model of unseen-class evaluation process. $D_L, D_U^S, D_U^U$, and $Acc$ denote labeled data, seen-class unlabeled data, unseen-class unlabeled data, and SSL model's performance of seen-class classification, respectively. The dashed line represents a confounding factor, which is the fundamental reason for the failure of previous evaluations.

classes on SSL model performance are flawed. Especially, previous safe SSL methods fix the size of the unlabeled dataset and adjust the proportion of unseen classes within the unlabeled data to assess the impact. This process contravenes the principle of controlling variables. Inspired by Peters et al. (2017), we utilize structural causal model to model the process in Fig. 1. As shown in Fig. 2(a), adjusting the proportion of unseen classes in unlabeled data alters the proportion of seen classes, meaning decreased classification performance of seen classes may not be due to an increase in unseen class samples in the unlabeled data, but rather a decrease in seen class samples. Thus, the prior flawed assessment standard that "unseen classes in unlabeled data can damage SSL model performance" may not always hold true.

To more reliably assess how unseen-class unlabeled data affects SSL model performance, removing the spurious correlation between seen-class unlabeled data $D_U^S$ and unseen-class unlabeled data $D_U^U$ is critical. Based on this, we propose the RE-SSL evaluation framework. Fig. 2(b) illustrates the dataset construction for our RE-SSL evaluation framework. $r_s$ denotes the ratio of selected seen-class data in $D_U^S$ to all seen-class data in $D_B^S$, that is $r_s = \frac{|D_U^S|}{|D_B^S|}$, where $|D_U^S|$ is the size of the sampled seen-class data, and $|D_B^S|$ is the size of the total set of seen-class data. $r_u$ denotes the ratio of selected unseen-class data in $D_U^U$ to all unseen-class data in $D_B^U$, that is $r_u = \frac{|D_U^U|}{|D_B^U|}$, where $|D_U^U|$ is the size of the sampled unseen-class data, and $|D_B^U|$ is the size of the total set of unseen-class data. In our assessment process, the size of the unlabeled dataset is no longer fixed; rather, the quantity of seen-class unlabeled data $D_U^S$ remains constant ($r_s$ is fixed) while only the quantity of unseen-class unlabeled data $D_U^U$ is varied ($r_u$ is variable).

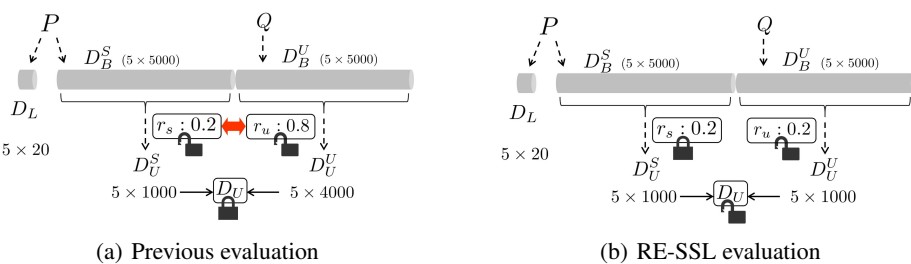

(a) Previous evaluation

(b) RE-SSL evaluation

Figure 2: An example of dataset construction in the previous evaluation and our RE-SSL evaluation framework. $r_s$ denotes the ratio of selected seen-class data to all seen-class data. $r_u$ denotes the ratio of selected unseen-class data to all unseen-class data. $P$ denotes the distribution of seen classes, and $Q$ denotes the distribution of unseen classes. $D_B^S$ and $D_B^U$ denote the initial set of seen classes and unseen classes, respectively. $D_L$ denotes the labeled set, $D_U$ denotes the unlabeled set. $D_U^S$ and $D_U^U$ together form $D_U$, where $D_U^S$ and $D_U^U$ are sampled from $D_B^S$ and $D_B^U$ according to $r_s$ and $r_u$, respectively. The red bidirectional arrow represents the confounding factor mentioned in Figure 1.

In addition to studying the impact of sample-number factor $r_u$ on SSL models, our RE-SSL evaluation framework also explored four additional settings with four factors, including category-number factor $C_n$, category-index factor $C_i$, nearness factor, and label distribution factor $C_{ib}$. Moreover, RE-SSL introduces five evaluation metrics, including the Slope of Regression function ($R_{slope}$), Global Magnitude (GM), Worst-case Adjacent Discrepancy (WAD), Best-case Adjacent Discrepancy (BAD), and Probability that $AD \geq 0$ ($P_{AD\geq0}$), and defined global and local robustness to assess the robustness of these algorithms against unseen classes. RE-SSL has conducted extensive experimental demonstrations and provided detailed analysis and insights.

**Our Contributions.** Our contributions contain five aspects: ① We first point out that the previous evaluations of the impact of unseen classes on SSL models are flawed, and construct a structural causal model to analyze the fundamental reasons leading to the flawed evaluations. ② We propose a new evaluation framework RE-SSL, which includes setting factors, constructing data, selecting comparative methods, formulating robustness evaluation metrics, and analyzing experimental results. ③ For the first time, we study the impact of unseen classes on SSL models from five perspectives, including unseen classes' sample-number, category-number, category-index, nearness, label distribution. ④ To effectively and intuitively analyze the impact of unseen classes on SSL models under different factors, we start from global and local robustness, proposing five metrics. ⑤ We conduct fair and reasonable experiments and provide detailed analysis based on the experimental results, offering strong guidance for SSL learning in scenarios where unseen classes appear.

## 2 FORMALIZATION OF SSL WITH UNSEEN CLASSES

Semi-supervsied learning harnesses the a large number of unlabeled data to alleviate reliance on limited labeled data. In a static environment, labeled data and unlabeled data are sampled from the identical distribution $P(\boldsymbol{x}, \boldsymbol{y})$, $\boldsymbol{x} \in \mathcal{X}$, $\boldsymbol{y} \in \mathcal{Y}$ on consistent data space $\mathcal{X} \times \mathcal{Y} \subseteq \mathbb{R}^d \times \{0, 1, \cdots, K-1\}$, where $d$ and $K$ denote the dimension of the input space and the number of classes. During training stage, we are given labeled set $D_L = \{(\boldsymbol{x}_i, y_i)|(\boldsymbol{x}_i, y_i) \sim P(\boldsymbol{x}, \boldsymbol{y})\}_{i=1}^{n_l}$ and unlabeled set $D_U = \{(\boldsymbol{x}_i)|\boldsymbol{x}_i \sim P(\boldsymbol{x})\}_{i=1}^{n_u}$, where $n_l$ and $n_u$ denote the number of samples in $D_L$ and $D_U$, $P(\boldsymbol{x})$ is the marginal distribution of $P(\boldsymbol{x}, \boldsymbol{y})$. Then, the goal of SSL is to learn a model with the smallest generalization error on $P(\boldsymbol{x}, \boldsymbol{y})$.

In a dynamic environment, labeled data and unlabeled data may not originate from the identical distribution. For example, unlabeled data may contain unseen classes which are not in labeled data. We assume that labeled set is sampled from the distribution $P(\boldsymbol{x}, \boldsymbol{y})$, and unlabeled set contains seen-class unlabeled set $D_U^S$ and unseen-class unlabeled set $D_U^U$, where $D_U^S$ are sampled from the distribution $P(\boldsymbol{x})$, but $D_U^U$ samples are from an additional distribution $Q(\boldsymbol{x})$, the label spaces of $P(\boldsymbol{x})$ and $Q(\boldsymbol{x})$ have no intersection.

## 3 RE-SSL EVALUATION FRAMEWORK

Previous most safe SSL works, such as DS3L Guo et al. (2020) and Safe-Student He et al. (2022), come to the conclusion: unseen-class unlabeled data impair the performance of SSL model. However, such a conclusion is based on an unfair evaluation process. Especially, as shown in Fig. 2(a), they fix the size of the unlabeled dataset $D_U$ and adjust the proportion of unseen classes within the unlabeled data to assess the impact. Adjusting the proportion $r_u$ of unseen classes in unlabeled data alters the proportion $r_s$ of seen classes, which results in the confounding factor between $D_U^S$ and $D_U^U$. The decreased classification performance of seen classes may not be due to an increase in unseen class samples in the unlabeled data, but rather a decrease in seen class samples. Thus, the impact of unseen-class unlabeled data on SSL models remains to be further verified. To evaluate the impact of unseen-class unlabeled data in a fair way, we propose the RE-SSL evaluation framework.

### 3.1 DATASET CONSTRUCTION

RE-SSL reconstructs the dataset used for evaluation, as shown in Fig 2(b). We only fix $r_s$ and no longer fix the size of $D_U$. By changing $r_u$ (abbreviated as $r$), we verify the impact of different numbers of unseen classes on the performance of SSL models. Furthermore, we also explore four additional settings: (1) varying the number of unseen class categories; (2) changing the indices of

unseen classes; (3) differing degrees of nearness in unseen classes; (4) diverse label distributions in unseen classes.

## 3.2 INTRODUCTION OF EVALUATION METRICS

Traditional SSL algorithms only consider the state where the unlabeled data distribution is consistent with the labeled data. Even most of the safe SSL methods (such as DS3L Guo et al. (2020) and Safe-Student He et al. (2022)) only consider one of all inconsistent states of unlabeled data and cannot directly and effectively reflect the impact of unseen classes on the performance of SSL models. Furthermore, previous safe SSL methods' analysis of unseen classes contradicts the standard controlled variable criterion. In fact, to study the impact of unseen classes on SSL models, it is necessary to strictly adhere to the controlled variable criterion. Currently, there is a lack of tools and evaluation metrics to directly and straightforwardly analyze the impact of unseen classes on the performance of SSL models from a dynamic, global perspective. Although some previous works in safe SSL (like DS3L and Safe-Student) have drawn curves describing the change in accuracy with the inconsistency of label space, they are based on an incorrect controlled variable criterion and have studied a limited range of inconsistency changes (such as 0-0.6). In addition, they lack a global metric to analyze the curves of different methods and cannot provide a direct comparison and analysis. To address the issues mentioned above, we propose $Acc$ Regression, which considers each state and its corresponding $Acc$ as points and performs linear regression to obtain the Regression function. This Regression function can globally measure the impact of unseen classes on different SSL methods.

To more comprehensively evaluate the impact of unseen classes on SSL models, we have defined multiple evaluation metrics to assess the robustness of these algorithms against changes in the label space distribution within unlabeled data. Unlike traditional SSL evaluations that solely focus on assessing $Acc(0)$ and secure SSL evaluations that only assess $Acc(r)$ for a specific $r$, our evaluation framework, meticulously based on the Regression function, allows for a more comprehensive and direct assessment of the impact of unlabeled data on SSL models.

**Slope of Regression function** ($R_{slope}$): The slope of Regression function reflects the overall trend of changes in SSL models as unseen classes are continuously added. We define Regression function as $Acc(r) = R_{slope} \cdot r + b$, where $b$ is just a constant. Then, based on $(r, \hat{Acc}(r)), r = 0, 0.2, 0.4, 0.5, 0.6, 0.8, 1.0$, where $\hat{ACC}$ represents the empirical accuracy corresponding to $r$, we use linear regression to obtain the regression function:

$$\mathrm{R}_{slope}(Acc) = \arg \min_{R_{slope}} \int_0^1 (Acc(r) - \hat{Acc}(r))^2 \, dr \tag{1}$$

**Global Magnitude (GM)**: Global magnitude reflects the influence of unseen classes on SSL models from a global perspective, which measures the integral of the magnitude of Acc for each $r$ to the expected accuracy $\bar{Acc}$, where $\bar{ACC}$ represents the mean of accuracy values of the model across all values of $r$. Unlike $R_{slope}$, this metric does not require imposing increasing or decreasing constraints on the $r$:

$$\mathrm{GM}(Acc) = \int_0^1 |\hat{Acc}(r) - \bar{Acc}(r)| \, dr \tag{2}$$

**Worst-case Adjacent Discrepancy (WAD)**:The maximum of adjacent discrepancy reflects the influence of unseen classes on SSL models from a local perspective, where adjacent discrepancy represents the local slope, calculating the change in $Acc$ between two adjacent $r$. WAD displays the worst-case impact of unseen classes have on SSL models:

$$\mathrm{WAD}(Acc) = \min \left\{ \frac{\hat{Acc}(r_{i+1}) - \hat{Acc}(r_i)}{r_{i+1} - r_i} \right\}_{i=1}^{n-1} \tag{3}$$

**Best-case Adjacent Discrepancy (BAD)**: The minimum of adjacent discrepancy reflects the influence of unseen classes on SSL models from a local perspective. BAD displays the best-case impact of unseen classes have on SSL models:

$$\mathrm{BAD}(Acc) = \max \left\{ \frac{\hat{Acc}(r_{i+1}) - \hat{Acc}(r_i)}{r_{i+1} - r_i} \right\}_{i=1}^{n-1} \tag{4}$$

Table 1: Evaluation on CIFAR10 with 100 labels under inconsistent label spaces.

| METHOD | r=0 | r=0.2 | r=0.4 | r=0.5 | r=0.6 | r=0.8 | r=1.0 | $R_{slope}$ | GM | BAD | WAD | $P_{AD\geq 0}$ |
|---|---|---|---|---|---|---|---|---|---|---|---|---|
| SUPERVISED | 0.617 | 0.617 | 0.617 | 0.617 | 0.617 | 0.617 | 0.617 | 0.000 | 0.000 | 0.000 | 0.000 | 1.000 |
| PSEUDOLABEL | 0.677 | 0.668 | 0.664 | 0.660 | 0.660 | 0.660 | 0.654 | -0.020 | 0.038 | **0.000** | -0.045 | 0.333 |
| PIMODEL | 0.667 | 0.651 | 0.644 | 0.637 | 0.636 | 0.636 | 0.630 | -0.034 | 0.066 | **0.000** | -0.080 | 0.167 |
| FIXMATCH | 0.612 | 0.547 | 0.515 | 0.514 | 0.505 | 0.442 | 0.365 | -0.223 | 0.386 | -0.010 | -0.385 | 0.000 |
| FLEXMATCH | 0.770 | 0.717 | 0.702 | 0.704 | 0.704 | 0.686 | 0.638 | -0.107 | 0.166 | 0.020 | -0.265 | 0.333 |
| MIXMATCH | 0.731 | 0.708 | 0.708 | 0.706 | 0.701 | 0.697 | 0.676 | -0.045 | 0.075 | **0.000** | -0.115 | 0.167 |
| UDA | 0.645 | 0.621 | 0.604 | 0.601 | 0.591 | 0.590 | 0.552 | -0.082 | 0.137 | -0.005 | -0.190 | 0.000 |
| SOFTMATCH | 0.764 | 0.710 | 0.689 | 0.697 | 0.692 | 0.682 | 0.607 | -0.112 | 0.193 | 0.080 | -0.375 | 0.167 |
| VAT | 0.710 | 0.677 | 0.663 | 0.663 | 0.657 | 0.651 | 0.639 | -0.063 | 0.111 | **0.000** | -0.165 | 0.167 |
| FREEMATCH | 0.760 | 0.723 | 0.665 | 0.635 | 0.608 | 0.605 | 0.440 | -0.287 | 0.496 | -0.015 | -0.825 | 0.000 |
| ICT | 0.618 | 0.619 | 0.621 | 0.620 | 0.621 | 0.619 | 0.621 | **0.002** | 0.007 | 0.010 | **-0.010** | **0.667** |
| UASD | 0.618 | 0.619 | 0.617 | 0.616 | 0.617 | 0.618 | 0.618 | 0.000 | **0.005** | 0.010 | **-0.010** | **0.667** |
| MTCF | 0.772 | 0.743 | 0.731 | 0.723 | 0.725 | 0.716 | 0.692 | -0.070 | 0.119 | 0.020 | -0.145 | 0.167 |
| CAFA | 0.652 | 0.652 | 0.640 | 0.653 | 0.641 | 0.642 | 0.640 | -0.013 | 0.040 | 0.130 | -0.120 | 0.500 |
| OPENMATCH | 0.713 | 0.606 | 0.586 | 0.595 | 0.579 | 0.517 | 0.473 | -0.211 | 0.350 | 0.090 | -0.535 | 0.167 |
| FIX_A_STEP | 0.662 | 0.634 | 0.615 | 0.582 | 0.555 | 0.585 | 0.509 | -0.138 | 0.272 | 0.150 | -0.380 | 0.167 |

**Probability that** $AD \geq 0$ **(**$P_{AD\geq 0}$**):** The probability that $AD \geq 0$ reflects the influence of unseen classes on SSL models from a global perspective. $P_{AD\geq 0}$ measures the probability that the addition of unseen classes does not decrease the performance of SSL models:

$$P_{AD\geq 0}(Acc) = \frac{1}{n}\sum_{i=1}^{n-1}\mathbb{I}(\hat{Acc}(r_{i+1}) - \hat{Acc}(r_i) \geq 0) \qquad (5)$$

Overall, we propose several targeted metrics from both global and local perspectives to assess the impact of unseen-class unlabeled data on SSL models. From a global perspective, we propose $R_{slope}$, GM, and $P_{AD\geq 0}$. From a local perspective, we introduce WAD and BAD. The integrals in Eq. 1 and Eq. 2 are computed as accumulation operations based on empirical values.

## 4 DEFINITION OF ROBUSTNESS TO UNSEEN CLASSES

Due to the existing definitions of robustness in previous work, to avoid confusion in the definition, we define global robustness (including $R_{slope}$ robustness) and local robustness (WAD robustness) based on $R_{slope}$ and WAD metrics.

In our safe SSL setting, if a SSL model does not exhibit a significant decline as unseen-class samples are continuously added, we consider the model to possess global robustness ($R_{slope}$ robustness) towards unseen classes. If, at any point during the ongoing addition of unseen-class samples, the performance of the SSL model does not decrease or even improves, we consider the model to have local robustness (WAD or BAD robustness) towards unseen classes.

**Definition 1** *(Global robustness.) SSL algorithm $\mathcal{A}$ returns a model $f_r \in \mathcal{F}$ using $D_L$ and $D_U$ with unseen-class ratio $r$ where $\mathcal{F}$ is the hypothesis space of $\mathcal{A}$. Let $Acc(r)$ denote the accuracy of $f_r$ on a test set with only seen classes. If there exists $\delta_g$ such that $R_{slope} \geq \delta_g$, we say that SSL algorithm $\mathcal{A}$ exhibits $\delta_g$-slope algorithmic robustness.*

**Definition 2** *(Local robustness.) If there exists $\delta_w$ such that $WAD \leq \delta_w$, we say that SSL algorithm $\mathcal{A}$ exhibits $\delta_w$-worst-case algorithmic robustness. If there exists $\delta_b$ such that $BAD \geq \delta_b$, we say that SSL algorithm $\mathcal{A}$ exhibits $\delta_b$-best-case algorithmic robustness.*

## 5 EXPERIMENTS

### 5.1 EVALUATED ALGORITHMS

Based on diversity and performance, we comprehensively consider 10 classical algorithms: PseudoLabel Lee et al. (2013), PiModel Laine & Aila (2017), ICT Verma et al. (2022), VAT Miyato et al. (2018), MixMatch Berthelot et al. (2019), UDA Xie et al. (2020), FixMatch Sohn et al. (2020), FlexMatch Zhang et al. (2021), FreeMatch Zhang et al. (2021) and SoftMatch Chen et al. (2023). We

Table 2: Evaluation on CIFAR100 with 1000 labels under inconsistent label spaces.

| METHOD | $r$=0 | $r$=0.2 | $r$=0.4 | $r$=0.5 | $r$=0.6 | $r$=0.8 | $r$=1.0 | $R_{slope}$ | GM | BAD | WAD | $P_{AD \geq 0}$ |
|---|---|---|---|---|---|---|---|---|---|---|---|---|
| SUPERVISED | 0.422 | 0.422 | 0.422 | 0.422 | 0.422 | 0.422 | 0.422 | 0.000 | 0.000 | 0.000 | 0.000 | 1.000 |
| PSEUDOLABEL | 0.421 | 0.417 | 0.413 | 0.411 | 0.417 | 0.415 | 0.413 | -0.006 | 0.018 | 0.060 | -0.020 | 0.167 |
| PIMODEL | 0.424 | 0.422 | 0.421 | 0.423 | 0.421 | 0.412 | 0.420 | -0.007 | 0.018 | 0.040 | -0.045 | 0.333 |
| FIXMATCH | 0.441 | 0.332 | 0.223 | 0.260 | 0.219 | 0.205 | 0.176 | -0.244 | 0.485 | **0.370** | -0.545 | 0.167 |
| FLEXMATCH | 0.443 | 0.412 | 0.390 | 0.377 | 0.382 | 0.344 | 0.317 | -0.120 | 0.208 | 0.050 | -0.190 | 0.167 |
| UDA | 0.380 | 0.383 | 0.380 | 0.386 | 0.388 | 0.388 | 0.396 | **0.015** | 0.029 | 0.060 | -0.015 | **0.833** |
| SOFTMATCH | 0.455 | 0.449 | 0.440 | 0.432 | 0.434 | 0.432 | 0.408 | -0.042 | 0.074 | 0.020 | -0.120 | 0.167 |
| VAT | 0.446 | 0.435 | 0.426 | 0.422 | 0.424 | 0.417 | 0.403 | -0.039 | 0.065 | 0.020 | -0.070 | 0.167 |
| FREEMATCH | 0.455 | 0.435 | 0.416 | 0.410 | 0.408 | 0.401 | 0.354 | -0.088 | 0.144 | -0.020 | -0.235 | 0.000 |
| ICT | 0.413 | 0.414 | 0.412 | 0.413 | 0.414 | 0.413 | 0.411 | -0.002 | **0.005** | 0.010 | **-0.010** | 0.500 |
| UASD | 0.415 | 0.411 | 0.411 | 0.411 | 0.415 | 0.412 | 0.411 | -0.002 | 0.011 | 0.040 | -0.020 | 0.500 |
| MTCF | 0.248 | 0.226 | 0.220 | 0.227 | 0.221 | 0.227 | 0.222 | -0.018 | 0.041 | 0.070 | -0.110 | 0.333 |
| CAFA | 0.424 | 0.423 | 0.414 | 0.418 | 0.418 | 0.419 | 0.412 | -0.010 | 0.022 | 0.040 | -0.045 | 0.500 |
| OPENMATCH | 0.367 | 0.329 | 0.325 | 0.309 | 0.324 | 0.307 | 0.292 | -0.063 | 0.115 | 0.150 | -0.190 | 0.167 |
| FIX_A_STEP | 0.448 | 0.393 | 0.359 | 0.342 | 0.326 | 0.311 | 0.241 | -0.188 | 0.326 | -0.075 | -0.350 | 0.000 |

also select 5 robust algorithms: UASD Chen et al. (2020), CAFA Huang et al. (2021), MTCF Yu et al. (2020), OpenMatch Saito et al. (2021) and Fix_A_Step Huang et al. (2023a).

## 5.2 EXPERIMANTAL SETTING

For all the experiments, we use supervised learning algorithm as baselines, which trains model based on $D_L$ with empirical risk minimization. The evaluation of the algorithm was performed based on the LAMDA-SSL toolkit Jia et al. (2022). We use ResNet50 He et al. (2016)as the backbone. All experiments of deep SSL algorithms are conducted with 4 NVIDIA RTX A6000 GPUs. To ensure reliability, we conduct three experiments for each sampling point with seed '0, 1, 2' to obtain average $Acc$. Different seeds are used to sample from the source dataset to create different training sets. Code is available at https://github.com/rundonghe/RESSL.

## 5.3 MAIN RESULTS AND ANALYSIS

Table 1 and 2 presents the evaluation of various SSL methods on CIFAR10 and CIFAR100, with assessments conducted using 100 labels within inconsistent label spaces and 1000 labels within inconsistent label spaces, respectively. The $r$ value indicates the degree of inconsistency within the label space, with $r = 0$ signifying complete consistency (i.e., no unseen classes) and $r = 1$ representing the highest degree of unseen class introduction. Performance metrics include accuracy (with values listed from $r = 0$ to $r = 1$), as well as $R_{slope}$, GM, WAD and BAD, and $P_{AD \geq 0}$. Based on Table 1, we can obtain the following conclusions:

**From the perspective of global robustness**: as unseen-class data is continuously integrated, the $Acc$ of the vast majority of SSL methods and robust SSL methods shows a declining trend. In CIFAR10, assume that $\sigma_g$ equals -0.020, according to Definition 1, several SSL methods PseudoLabel and ICT, as well as robust SSL methods UASD and CAFA, which are relatively robust to unseen classes, the addition of unseen classes essentially has no impact on their algorithmic performance. On the contrary, some SSL methods such as FixMatch and FreeMatch, as well as robust SSL methods like OpenMatch, are very sensitive to unseen classes, with their $R_{slope}$ values being -0.223, -0.287, and -0.211 respectively. The introduction of unseen classes can severely deteriorate their performance. In CIFAR100, assume that $\sigma_g$ also equals -0.020, according to Definition 1, several SSL methods PseudoLabel, PiModel, UDA and ICT, as well as robust SSL methods UASD, MTCF, and CAFA, which are relatively robust to unseen classes. Then, examining both CIFAR10 and CIFAR100, we find that the SSL algorithms PseudoLabel and ICT, as well as the robust SSL algorithms UASD and CAFA, are robust to unseen classes in both CIFAR10 and CIFAR100. This also reflects the stability of these algorithms' robustness to unseen classes, unaffected by the choice of dataset.

**From the perspective of local robustness**: In the context of local robustness, an analysis of SSL algorithms on CIFAR10 and CIFAR100 datasets centers on the performance metrics of Worst-case Adjacent Discrepancy (WAD) and Best-case Adjacent Discrepancy (BAD). For the CIFAR10 dataset, the analysis of WAD reveals the extent of performance degradation under locally worst-case scenarios. For instance, FixMatch exhibits a significant drop (WAD at -0.385), indicating weaker

Table 3: Evaluation on CIFAR10 with 100 labels under the different number of unseen classes.

| METHOD | base | $C_n$=1 | $C_n$=2 | $C_n$=3 | $C_n$=4 | $C_n$=5 | $R_{slope}$ | GM | BAD | WAD | $P_{AD \geq 0}$ |
|---|---|---|---|---|---|---|---|---|---|---|---|
| SUPERVISED | 0.617 | 0.617 | 0.617 | 0.617 | 0.617 | 0.617 | 0.000 | 0.000 | 0.000 | 0.000 | 1.000 |
| PSEUDOLABEL | 0.677 | 0.648 | 0.652 | 0.663 | 0.664 | 0.668 | 0.005 | 0.036 | 0.011 | 0.001 | **1.000** |
| PIMODEL | 0.667 | 0.622 | 0.628 | 0.640 | 0.642 | 0.651 | 0.007 | 0.046 | 0.012 | **0.002** | **1.000** |
| FIXMATCH | 0.612 | 0.710 | 0.674 | 0.574 | 0.441 | 0.547 | -0.056 | 0.411 | **0.106** | -0.133 | 0.250 |
| FLEXMATCH | 0.770 | 0.658 | 0.644 | 0.689 | 0.699 | 0.715 | 0.017 | 0.120 | 0.045 | -0.014 | 0.750 |
| UDA | 0.645 | 0.620 | 0.599 | 0.594 | 0.621 | 0.621 | 0.002 | 0.058 | 0.027 | -0.021 | 0.500 |
| SOFTMATCH | 0.764 | 0.671 | 0.662 | 0.724 | 0.715 | 0.710 | 0.013 | 0.119 | 0.062 | -0.009 | 0.250 |
| VAT | 0.710 | 0.606 | 0.627 | 0.671 | 0.676 | 0.677 | **0.019** | 0.139 | 0.044 | 0.001 | **1.000** |
| FREEMATCH | 0.760 | 0.668 | 0.639 | 0.724 | 0.688 | 0.723 | 0.016 | 0.140 | 0.085 | -0.036 | 0.500 |
| ICT | 0.618 | 0.611 | 0.613 | 0.615 | 0.619 | 0.619 | 0.002 | 0.014 | 0.004 | 0.000 | **1.000** |
| UASD | 0.618 | 0.618 | 0.620 | 0.617 | 0.619 | 0.619 | 0.000 | **0.004** | 0.002 | -0.003 | 0.750 |
| MTCF | 0.772 | 0.723 | 0.704 | 0.737 | 0.736 | 0.743 | 0.007 | 0.060 | 0.033 | -0.019 | 0.500 |
| CAFA | 0.652 | 0.642 | 0.645 | 0.642 | 0.643 | 0.648 | 0.001 | 0.010 | 0.005 | -0.003 | 0.750 |
| OPENMATCH | 0.713 | 0.642 | 0.634 | 0.633 | 0.561 | 0.606 | -0.015 | 0.127 | 0.045 | -0.072 | 0.250 |
| FIX_A_STEP | 0.662 | 0.674 | 0.647 | 0.631 | 0.625 | 0.632 | -0.011 | 0.075 | 0.007 | -0.027 | 0.250 |

Table 4: Evaluation on CIFAR10 with 100 labels under the different index of unseen classes.

| METHOD | | | | near | | | | | | far | | | |
|---|---|---|---|---|---|---|---|---|---|---|---|---|---|
| | base | $C_i$=5 | $C_i$=6 | $C_i$=7 | $C_i$=8 | $C_i$=9 | GM | $C_i$=5 | $C_i$=6 | $C_i$=7 | $C_i$=8 | $C_i$=9 | GM |
| SUPERVISED | 0.617 | 0.617 | 0.617 | 0.617 | 0.617 | 0.617 | 0.000 | 0.617 | 0.617 | 0.617 | 0.617 | 0.617 | 0.000 |
| PSEUDOLABEL | 0.677 | 0.648 | 0.649 | 0.650 | 0.650 | 0.648 | **0.004** | 0.614 | 0.618 | 0.626 | 0.621 | 0.623 | 0.018 |
| PIMODEL | 0.667 | 0.638 | 0.642 | 0.639 | 0.637 | 0.622 | 0.027 | 0.621 | 0.624 | 0.625 | 0.621 | 0.624 | 0.008 |
| FIXMATCH | 0.612 | 0.635 | 0.504 | 0.533 | 0.527 | 0.710 | 0.363 | 0.704 | 0.694 | 0.689 | 0.695 | 0.702 | 0.025 |
| FLEXMATCH | 0.770 | 0.656 | 0.640 | 0.682 | 0.662 | 0.658 | 0.050 | 0.610 | 0.586 | 0.619 | 0.606 | 0.616 | 0.046 |
| UDA | 0.645 | 0.627 | 0.646 | 0.627 | 0.608 | 0.620 | 0.046 | 0.644 | 0.640 | 0.640 | 0.643 | 0.633 | 0.014 |
| SOFTMATCH | 0.764 | 0.654 | 0.665 | 0.725 | 0.687 | 0.671 | 0.102 | 0.646 | 0.650 | 0.639 | 0.644 | 0.642 | 0.015 |
| VAT | 0.710 | 0.654 | 0.632 | 0.657 | 0.623 | 0.606 | 0.084 | 0.564 | 0.565 | 0.577 | 0.568 | 0.581 | 0.032 |
| FREEMATCH | 0.760 | 0.655 | 0.610 | 0.709 | 0.628 | 0.668 | 0.140 | 0.642 | 0.633 | 0.641 | 0.639 | 0.637 | 0.014 |
| ICT | 0.618 | 0.623 | 0.619 | 0.622 | 0.608 | 0.611 | 0.028 | 0.597 | 0.596 | 0.598 | 0.597 | 0.597 | **0.002** |
| UASD | 0.618 | 0.617 | 0.616 | 0.618 | 0.619 | 0.618 | **0.004** | 0.617 | 0.618 | 0.618 | 0.618 | 0.617 | **0.002** |
| MTCF | 0.772 | 0.703 | 0.685 | 0.697 | 0.689 | 0.723 | 0.054 | 0.632 | 0.573 | 0.619 | 0.596 | 0.596 | 0.089 |
| CAFA | 0.652 | 0.639 | 0.639 | 0.638 | 0.634 | 0.630 | 0.016 | 0.606 | 0.597 | 0.592 | 0.612 | 0.613 | 0.038 |
| OPENMATCH | 0.713 | 0.636 | 0.546 | 0.648 | 0.566 | 0.642 | 0.206 | 0.607 | 0.559 | 0.569 | 0.629 | 0.622 | 0.133 |
| FIX_A_STEP | 0.662 | 0.630 | 0.630 | 0.638 | 0.634 | 0.630 | 0.014 | 0.693 | 0.684 | 0.693 | 0.691 | 0.693 | 0.014 |

local robustness against unseen classes. In contrast, the ICT algorithm's WAD value is nearly zero (0.010), demonstrating stronger local worst-case robustness. Moreover, BAD values suggest that FlexMatch (0.166) and VAT (0.111) possess higher local best-case robustness, implying that these algorithms can capitalize on favorable conditions to maximize performance enhancements. Moving to the CIFAR100 dataset, we observe universally lower WAD values, particularly for FixMatch (-0.545), which shows high sensitivity to unseen classes and insufficient local worst-case robustness. Concurrently, FlexMatch shows a significant advantage in local best-case robustness with a BAD value of 0.208, while Fix_A_STEP's BAD value (-0.350) indicates a lower capacity for performance improvement under optimal conditions. In summary, regarding local robustness, FlexMatch and ICT display relatively better performance stability, both under worst-case scenarios and best-case scenarios. On the contrary, FixMatch shows deficiencies in both aspects. These insights are crucial for the design and deployment of SSL algorithms that are robust in dynamic environments.

## 5.4 EXTENDED EXPERIMENTS

Before this, our research had only examined the impact of a single element (the number of unseen class samples) on SSL models. However, in more complex scenarios, unseen class samples may have varying numbers of unseen class categories, unseen class samples may have different unseen class category indices, unseen class samples may have varying degrees of nearness, and unseen class samples may have different labeling distributions. Studying these elements is also of significant instructive value for SSL models.

**Category-number factor** $C_n$**:** $C_n$ refers to the factor that controls the number of unseen-class categories from which the unseen class samples originate. We validated different approaches on

Table 5: Evaluation on CIFAR10 with 100 labels under different imbalance factors of unseen classes.

| METHOD | base | $C_{ib}$=0.01 | $C_{ib}$=0.02 | $C_{ib}$=0.05 | $C_{ib}$=0.10 | $C_{ib}$=0.20 | $R_{slope}$ | GM | BAD | WAD | $P_{AD \geq 0}$ |
|---|---|---|---|---|---|---|---|---|---|---|---|
| SUPERVISED | 0.617 | 0.617 | 0.617 | 0.617 | 0.617 | 0.617 | 0.000 | 0.000 | 0.000 | 0.000 | 1.000 |
| PSEUDOLABEL | 0.677 | 0.660 | 0.660 | 0.663 | 0.658 | 0.662 | 0.005 | 0.008 | 0.100 | -0.100 | **0.750** |
| PIMODEL | 0.667 | 0.647 | 0.649 | 0.642 | 0.646 | 0.642 | -0.026 | 0.013 | 0.200 | -0.233 | 0.500 |
| FIXMATCH | 0.612 | 0.529 | 0.534 | 0.541 | 0.603 | 0.560 | **0.208** | 0.112 | 1.240 | -0.430 | **0.750** |
| FLEXMATCH | 0.770 | 0.710 | 0.703 | 0.714 | 0.706 | 0.717 | 0.044 | 0.022 | 0.367 | -0.700 | 0.500 |
| UDA | 0.645 | 0.620 | 0.647 | 0.634 | 0.615 | 0.619 | -0.088 | 0.054 | **2.700** | -0.433 | 0.500 |
| SOFTMATCH | 0.764 | 0.730 | 0.731 | 0.725 | 0.721 | 0.698 | -0.170 | 0.046 | 0.100 | -0.230 | 0.250 |
| VAT | 0.710 | 0.681 | 0.682 | 0.682 | 0.678 | 0.675 | -0.037 | 0.012 | 0.100 | -0.080 | 0.500 |
| FREEMATCH | 0.760 | 0.730 | 0.723 | 0.674 | 0.670 | 0.677 | -0.256 | 0.127 | 0.070 | -1.633 | 0.250 |
| ICT | 0.618 | 0.622 | 0.621 | 0.621 | 0.622 | 0.622 | 0.003 | **0.002** | 0.020 | -0.100 | **0.750** |
| UASD | 0.618 | 0.618 | 0.618 | 0.617 | 0.618 | 0.617 | -0.004 | **0.002** | 0.020 | **-0.033** | 0.500 |
| MTCF | 0.772 | 0.722 | 0.720 | 0.722 | 0.721 | 0.722 | 0.004 | 0.004 | 0.067 | -0.200 | 0.500 |
| CAFA | 0.652 | 0.647 | 0.650 | 0.643 | 0.648 | 0.646 | -0.007 | 0.009 | 0.300 | -0.233 | 0.500 |
| OPENMATCH | 0.713 | 0.613 | 0.613 | 0.620 | 0.650 | 0.639 | 0.163 | 0.070 | 0.600 | -0.110 | **0.750** |
| FIX_A_STEP | 0.622 | 0.685 | 0.638 | 0.632 | 0.633 | 0.628 | -0.179 | 0.084 | 0.020 | -4.700 | 0.250 |

CIFAR10, with the results shown in Table 3. Here, $base$ refers to the scenario where $r$=0, meaning no unseen class data is added. In other cases, $r$ is set to 0.2. On CIFAR10, $C_n$ is set to 1, 2, 3, 4, 5.

On the CIFAR10 dataset, most methods show high accuracy under the $base$ scenario, and the accuracy generally rises as the number of unseen classes $C_n$ increases. The $R_{slope}$ column indicates that the impact on the SSL model is continuously mitigated for most methods with the increase in the number of unseen class categories, except for FixMatch, OpenMatch, and Fix_A_Step. Specifically, FixMatch shows a sharp decline in accuracy at $C_n = 4$, suggesting that it may need additional improvements when dealing with higher category diversity.

**Category-index factor $C_i$:** $C_i$ refers to the factor that controls the index of unseen-class category from which the unseen class samples originate. We validated different approaches on CIFAR10 and the results can be seen in Table 4 with near part. Unlike the previous $r$ and $C_n$, the different values of $C_i$ do not have a progressive relationship, so we can only use GM to measure the impact of $C_i$ on the SSL model. Because we select the first five classes as seen classes in CIFAR10, the range of $C_i$ is 5, 6, 7, 8, 9. $base$ still refers to the scenario where $r$=0, meaning no unseen class data is added.

Almost all methods perform better under the base scenario than when unseen classes with different indices are added. As $C_i$ varies from 5 to 9, the accuracy trends are inconsistent, reflecting the varied impact of unseen class indices on different methods. For instance, FixMatch shows a significantly low accuracy at $C_i = 6$ (0.504), but its performance recovers and even exceeds the base level when $C_i$ is at 5 or other values. This indicates that unseen class data with different indices can have a large and variable impact on SSL models, potentially improving or degrading their performance. PseudoLabel and UASD have very low GM values, suggesting they maintain better robustness across different unseen class indices. On the other hand, FixMatch and OpenMatch have high GM values, indicating that these two methods are quite sensitive to $C_i$.

**Nearness factor:** Prior to this, the unseen classes in our experiments originated from near out-of-distribution (OOD). For instance, in our experiments on CIFAR10, we default to treating the first five classes as seen and the latter five as unseen. However, since they all come from the same dataset, there is a considerable degree of relatedness in terms of data categories and distribution. But what if the nearness of the unseen classes is different, such as when the unseen classes come from far OOD? How would these unseen classes affect the SSL models? To investigate the impact of nearness on SSL models, we have also conducted experiments where the unseen classes come from MNIST, based on the previous CIFAR10 experiments, to make horizontal and vertical comparisons. The experimental results can be seen in Table 4 with far part.

When comparing the "near" and "far" results, it's apparent that almost all SSL models have a notably lower GM in the far scenario than in the near one, indicating that the SSL models are more robust to the Category-index factor $C_i$ in the far scenario. Moreover, we find that except for FixMatch and Fix_A_Step, the performance of the majority of SSL models under the same settings is better in the near scenario than in the far one. This suggests that the smaller the semantic shift and distribution shift from the seen classes to the unseen classes, the less damage to the SSL models. This conclusion is clearly valid. For example, when mixup is applied directly to the seen classes as unseen classes, at

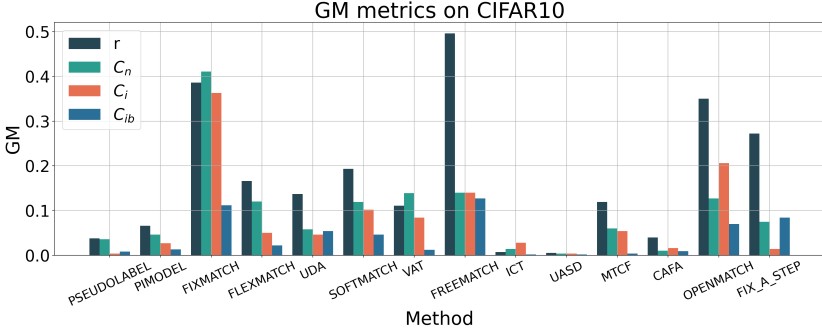

Figure 3: GM metrics on CIFAR10 with 100 labels under different factors.

this time these unseen classes have the smallest semantic shift and distribution shift from the seen classes. The classes obtained after mixup are clearly helpful for the classification of seen classes.

**Label distributions factor $C_{ib}$:** $C_{ib}$ refers to the factor that controls the labeling distribution of the unseen class data. Specifically, we achieve varying degrees of class imbalance by adjusting the sampling strategy of the samples. We set the imbalance factor ($C_{ib}$) to 0.01, 0.02, 0.05, 0.10, and 0.20. The smaller the $C_{ib}$, a higher degree of class imbalance.

Through Table 5, we can observe that different SSL methods react differently to changes in $C_{ib}$. Most methods are quite robust to $C_{ib}$, with R_slope and GM values close to 0. However, FixMatch and OpenMatch show a generally increasing trend as $C_{ib}$ increases. Similarly, FreeMatch and Fix_A_Step also tend to exhibit an increasing trend with the increase in $C_{ib}$.

## 6 ANALYSIS OF EXPERIMENTAL RESULTS

### 6.1 ANALYSIS OF FACTORS

Table 6 and Fig. 3 provides the GM values for various SSL methods on the CIFAR10 dataset with 100 labels under different factors. These factors include the number of unseen class samples ($r$), the number of unseen class categories ($C_n$), the index of unseen class categories ($C_i$), and the label distribution of unseen classes ($C_{ib}$). $F_{avg}$ represents the mean of each row, while $A_{avg}$ represents the mean of each column. As for the degree of nearness of unseen classes, since we only verified two scenarios of nearness for unseen classes, namely near OOD and far OOD, we are unable to obtain a GM metrics for the degree of nearness of unseen classes. In this table, the GM metric is used to evaluate the robustness of various SSL methods under the influence of different factors. The lower the GM metrics, the less sensitive the model is to that specific factor, indicating greater robustness. According to Table 6, the average GM value for $r$ is the highest at 0.170, indicating that SSL models are generally sensitive

Table 6: GM under different factors.

| Method | $r$ | $C_n$ | $C_i$ | $C_{ib}$ | $F_{avg}$ |
|---|---|---|---|---|---|
| PseudoLabel | 0.038 | 0.036 | 0.004 | 0.008 | 0.021 |
| PiModel | 0.066 | 0.046 | 0.027 | 0.013 | 0.038 |
| FixMatch | 0.386 | 0.411 | 0.363 | 0.112 | 0.318 |
| FlexMatch | 0.166 | 0.120 | 0.050 | 0.022 | 0.089 |
| UDA | 0.137 | 0.058 | 0.046 | 0.054 | 0.073 |
| SoftMatch | 0.193 | 0.119 | 0.102 | 0.046 | 0.115 |
| VAT | 0.111 | 0.139 | 0.084 | 0.012 | 0.086 |
| FreeMatch | 0.496 | 0.140 | 0.140 | 0.127 | 0.225 |
| ICT | 0.007 | 0.014 | 0.028 | 0.002 | 0.012 |
| UASD | 0.005 | 0.004 | 0.004 | 0.002 | 0.003 |
| MTCF | 0.119 | 0.060 | 0.054 | 0.004 | 0.059 |
| CAFA | 0.040 | 0.010 | 0.016 | 0.009 | 0.018 |
| OpenMatch | 0.350 | 0.127 | 0.206 | 0.070 | 0.188 |
| Fix_A_Step | 0.272 | 0.075 | 0.014 | 0.084 | 0.111 |
| $A_{avg}$ | 0.170 | 0.097 | 0.081 | 0.040 | - |

to changes in the number of unseen classes in the unlabeled sample set. Methods like FreeMatch, with a GM value of 0.496, are particularly sensitive to this factor. The average GM value for $C_n$ is 0.097, which, although lower than the impact of $r$, still indicates a certain level of sensitivity. This suggests that altering the number of unseen classes can also affect the robustness of the model. The average GM value for $C_i$ is 0.081, showing that models also have a certain sensitivity to changes in the index of unseen classes, but this sensitivity is less than the impact of changes in the number of unseen class samples and the number of unseen class categories. The average GM value for $C_{ib}$ is

the lowest at 0.040, meaning that the models are relatively robust to changes in the label distribution of unseen classes. This indicates that changes in the label distribution of unseen classes have a less significant impact on model performance compared to other factors. Overall, these results reveal that SSL learning models are most sensitive to the quantity of unseen class samples when dealing with unseen categories, while being relatively robust to changes in the label distribution of unseen classes. This insight helps us understand how models perform under different types of data uncertainty.

## 6.2 ANALYSIS OF ALGORITHMS

Based on Table 6, we can analyze the overall differences between various SSL methods. The smaller the $F_{avg}$, the more robust the algorithm is to unseen classes. PseudoLabel and PiModel have a very small $F_{avg}$, indicating that they are quite robust to unseen classes. PseudoLabel relies solely on the inconsistency between soft and hard labels, while PiModel utilizes the inconsistency caused by weak data augmentation. The unsupervised loss derived from their inconsistencies is minimal, hence the algorithms exhibit strong robustness. ICT's $F_{avg}$ is very small, even smaller than that of PseudoLabel and PiModel. This is because ICT employs Mixup operations between labeled and unlabeled data, which can effectively mitigate the impact of unseen classes in the unlabeled data on the classification of seen classes, thus ensuring the robustness of ICT to unseen classes. In classic SSL algorithms, FixMatch has the largest $F_{avg}$ of 0.318. This demonstrates that FixMatch is the most sensitive to unseen classes and is not robust enough. We analyze this as being due to FixMatch using a fixed threshold, which is not suitable for complex scenarios. Methods that are improvements based on FixMatch, such as FlexMatch, FreeMatch, and SoftMatch, have abandoned the use of fixed thresholds and instead employ adaptive thresholds or appropriate weighting strategies. Therefore, they have achieved a certain degree of enhancement in robustness.

As for safe SSL models, UASD exhibits the strongest robustness. We analyze that this is because it leverages the inconsistency between predictions from the current and previous rounds to update the model, thus exhibiting greater robustness to unseen classes. CAFA, which is designed based on PiModel and Mixup—both inherently more robust to unseen classes—also demonstrates enhanced robustness to unseen classes. MTCF employs curriculum learning and multi-task learning to filter out unseen classes, therefore also showing a more robust nature to unseen classes. OpenMatch, while relatively less robust, is based on FixMatch design, but is more robust than FixMatch because it introduces one-vs-all (OVA) classifiers, which can help filter out unseen classes. Similarly, Fix-A-Step also shows relatively less robustness as it is based on the FixMatch design. However, it is more robust than FixMatch because it selectively discards the unlabeled loss by calculating the cosine similarity of the gradients of the labeled and unlabeled losses, which is a very interesting and promising approach.

## 7 LIMITATION

This manuscript primarily explores the impact of unseen classes within unlabeled data on semi-supervised models. However, in practical applications, the appearance of unseen classes in unlabeled data is not the only change; seen classes in unlabeled data may also experience shifts in feature and label distributions. Investigating the influence of unseen classes on semi-supervised models in more complex scenarios is also crucial. Additionally, another weakness of our manuscript is the lack of corresponding theoretical analysis.

## 8 CONCLUSION

This article illuminates the deficiencies in prior assessments concerning the influence of unseen classes on SSL models, and pioneers a structural causal model to dissect the underlying causes of these evaluation inaccuracies. We introduce the RE-SSL framework, a holistic approach encompassing factor specification, dataset structuring, comparative methodology, development of robustness metrics, and comprehensive experimental analysis. This study marks the inaugural investigation into the effects of unseen classes on SSL models across five critical dimensions. By leveraging both global and local robustness metrics, we offer a nuanced and intuitive examination of the unseen classes' impact under varied conditions. Our rigorous and methodologically sound experiments, coupled with an in-depth analysis, pave the way for significant advancements in SSL, particularly in environments featuring unseen classes, thereby setting new standards for model robustness.

# 9 ACKNOWLEDGMENT

This research was supported by Westlake University Research Center for Industries of the Future, Westlake University Center for High-performance Computing, the National Natural Science Foundation of China (Grant No. U23A20389, 62176139, 62306133). The content is solely the responsibility of the authors and does not necessarily represent the official views of the funding entities. We would like to thank the reviewers for their constructive suggestions.

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

## A    RELATED WORK

### A.1    DEEP SEMI-SUPERVISED LEARNING

Semi-supervised learning (SSL) holds immense potential within the realm of machine learning Chapelle et al. (2009); Van Engelen & Hoos (2020); Wang et al. (2022b); Yang et al. (2023; 2022); Zheng et al. (2022). Compared to supervised learning, it escapes the shackles of being entirely reliant on labeled data, shifting its focus towards the rich sample characteristics and structural information concealed within unlabeled data Chen et al. (2023). In contrast with unsupervised learning, it successfully propagates the semantic information from labeled data to unlabeled counterparts, thereby streamlining its application in downstream tasks.

Self-training and consistency regularization stand out as cornerstone techniques of contemporary deep SSL Laine & Aila (2017); Li et al. (2021; 2023); Sajjadi et al. (2016); Tarvainen & Valpola (2017); Wang et al. (2023). In particular, the pseudo-labeling technique within self-training plays a pivotal role. It iteratively enlarges the "labeled data" by speculatively assigning pseudo-labels to unlabeled samples, enabling unlabeled data to generate supervisory signals for model training Lee et al. (2013); Wang et al. (2022a). This strategy has become almost inescapable in current SSL paradigms. Consistency regularization, on the other hand, operates under the presumption that training samples share identical labels with their proximate synthetic counterparts (or similar views). By disseminating the labels from seen-class samples and confident predictions of unseen-class samples, it harnesses unlabeled data even more effectively Berthelot et al. (2019); Miyato et al. (2018); Saito et al. (2021); Sohn et al. (2020); Huang et al. (2023b); Guo et al. (2025). For example, FixMatch Sohn et al. (2020) is an efficient approach that capitalizes on both techniques. It refines consistency regularization through the distinct deployment of weak and strong augmentations. And there are many other works that have made important technical contributions to the research of SSL. ReMixMatch Kurakin et al. (2020) introduces distribution alignment and augmentation anchoring. FlexMatch Zhang et al. (2021) and FreeMatch Wang et al. (2022c) propose to introduce currency pseudo labeling and self-adaptive threshold to deal with the learning difficulty of different classes.

Despite the promising strides in SSL, there are inherent limitations that need to be addressed. The most salient among these is the prevalent assumption that labeled and unlabeled data samples share identical category spaces. When this assumption is breached, there may be a tangible dip in the performance of seen-class classification.

### A.2    SAFE SEMI-SUPERVISED LEARNING

To address the issues mentioned above, mainstream safe SSL strategies tend to detect and exclude unseen-class samples to avoid misunderstandings in the training process. For example, UASD Chen et al. (2020) relies on the confidence threshold of model predictions to identify outliers. DS3L Guo et al. (2020) assigns soft weights to each unlabeled instance by a weighting function based on bi-level optimization. Safe-Student He et al. (2022) utilizes the logit output of the model to define an energy difference score. Furthermore, methods like MTCF Yu et al. (2020), CAFA Huang et al. (2021),

OpenMatch Saito et al. (2021), and Fix_A_Step Huang et al. (2023a) have introduced specific modules dedicated to the detection of unseen-classes.

However, previous methods for assessing the impact of unseen classes on SSL model performance are flawed. They fix the size of the unlabeled dataset and adjust the proportion of unseen classes within the unlabeled data to assess the impact. This process contravenes the principle of controlling variables. Adjusting the proportion of unseen classes in unlabeled data alters the proportion of seen classes, meaning the decreased classification performance of seen classes may not be due to an increase in unseen class samples in the unlabeled data, but rather a decrease in seen class samples. Thus, the prior flawed assessment standard that "unseen classes in unlabeled data can damage SSL model performance" may not always hold true.

## B  FURTHER INTRODUCTION

### B.1  INTRODUCTION OF COMPARED SSL METHODS

We conduct experiments on 10 popular SSL methods, 5 safe SSL methods. The 10 SSL methods we employed can be divided into three categories. **Pseudo-Labeling Methods:** This category involves generating pseudo-labels for unlabeled data and using these labels to guide model training. PseudoLabel Lee et al. (2013) assigns the highest confidence class prediction as the pseudo-label. FixMatch Sohn et al. (2020) further refines this approach by training only on pseudo-labels exceeding a certain confidence threshold, combined with data augmentation to enhance label quality. FlexMatch Zhang et al. (2021) and FreeMatch Zhang et al. (2021) dynamically adjust thresholds based on the learning difficulty of different classes to balance learning progress. SoftMatch Chen et al. (2023) optimizes the quality and quantity of pseudo-labels by weighting them with a truncated Gaussian function based on sample confidence. **Consistency Regularization Methods:** These methods emphasize the importance of the model maintaining consistent outputs in response to minor perturbations of input data. PiModel Laine & Aila (2017) applies data and model perturbations to the same unlabeled sample, defining consistency loss through the mean squared error (MSE) between two outputs. MixMatch Berthelot et al. (2019) employs the MixUp Zhang et al. (2017) technique on both labeled and unlabeled data, creating new training data by mixing samples and their labels to calculate consistency loss for improved model generalization. UDA Xie et al. (2020) introduces perturbations with advanced data augmentation techniques and uses Kullback-Leibler (KL) divergence as consistency loss to ensure output consistency. ICT Verma et al. (2022) ensures prediction consistency for interpolated samples with the original samples' prediction interpolations. **Adversarial Training Methods:** Represented by VAT Miyato et al. (2018), this approach defines consistency loss by identifying the virtual adversarial direction that maximizes the perturbation in model output without relying on label information. This method aims to improve the model's robustness to minor variations in input data, thereby enhancing its generalization capability.

In addressing the challenges of SSL, researchers have proposed various safe SSL methods designed to effectively utilize unlabeled data while identifying and filtering out Out-Of-Distribution (OOD) samples. UASD Chen et al. (2020) derives ensemble predictions by averaging all previous network predictions, conducting uncertainty-aware self-distillation based on these ensemble predictions. MTCF Yu et al. (2020) initializes the OOD scores for both unlabeled and labeled samples and alternates between updating the network parameters and OOD scores, cleansing the noisy OOD scores of unlabeled samples in a joint optimization process. CAFA Huang et al. (2021) introduces a scoring mechanism to identify both labeled and unlabeled data within shared categories and employs domain adaptation to increase the efficiency of unlabeled data utilization. OpenMatch Saito et al. (2021) combines FixMatch with novel detection techniques based on one-vs-all classifiers, introducing an open-set soft-consistency regularization loss to enhance outlier detection capabilities. Fix_A_Step Huang et al. (2023a) integrates MixUp with step direction modification strategies, optimizing the model's ability to utilize unlabeled data while maintaining the accuracy of the labeled set by adjusting the relationship between labeled and unlabeled data from a gradient perspective.

### B.2  FURTHER STATEMENT ON EVALUATION FRAMEWORK

In real-world applications, the primary focus is often on model performance with a fixed size of unlabeled data, rather than on how performance changes as the proportion of unseen classes ($r$) varies.

Fixing the size of the unlabeled set is a practical approach, as the exact proportion of unseen classes is typically unknown during data collection. This framework provides valuable insights for practical use cases.

However, evaluating semi-supervised learning models under a single fixed $r$ value may not fully capture their robustness or generalizability. Evaluating performance across multiple $r$ values offers a comprehensive understanding of model behavior under diverse conditions. This approach enables a systematic assessment of how the proportion of unseen classes impacts performance, highlighting robustness and adaptability.

Both evaluation approaches address distinct aspects of semi-supervised learning. Fixing the size of the unlabeled set emphasizes practical relevance for real-world scenarios, while evaluating varying proportions of unseen classes provides insights into model robustness and adaptability. Together, these perspectives contribute to a more holistic understanding of semi-supervised learning models.

## C  EXPERIMENTS

### C.1  EVALUATION METRIC

Table 7 shows evaluation metric with range, interpretation of positive/negative values, and how it relates to model performance.

### C.2  EXPERIMENTAL DETAIL

#### C.2.1  REGARDING THE NUMBER OF UNSEEN-CLASS EXAMPLES

For CIFAR-10, we select the first 5 classes as seen classes and the last 5 classes as unseen classes. From the training set of known classes (which contains 25,000 samples), we randomly select 100 samples to form the labeled set $D_L$. We then randomly select $r_s \times 25,000$ samples from the remaining samples of the training set of known classes and put them into $D_U^S$. From the training set of unknown classes (which contains 25,000 samples), we randomly select $r_u \times 25,000$ samples and put them into $D_U^U$. $D_U^S$ and $D_U^U$ together form the unlabeled set $D_U$.

For CIFAR-100, we select the first 50 classes as seen classes and the last 50 classes as unseen classes. Other settings are the same as CIFAR-10.

#### C.2.2  REGARDING THE NUMBER OF UNSEEN-CLASS CATEGORIES

We fix the number of unseen-class examples by setting $r = 0.2$. Unlike the first setup, instead of randomly selecting 25,000 samples from the entire training set of unknown classes, we randomly select 25,000 samples from the last $C_n$ classes of the unknown classes and put them into $D_U^U$.

#### C.2.3  REGARDING THE INDICES OF UNSEEN CLASSES

We fix the number of unseen-class examples by setting $r = 0.2$. Unlike the first and second setups, we randomly select $r \times 25,000$ samples from the $C_i$-th class of the unknown classes and put them into $D_U^U$.

#### C.2.4  REGARDING THE DEGREES OF NEARNESS OF UNSEEN CLASSES

Similar to the third setup, instead of randomly selecting $r \times 25,000$ samples from the $C_i$-th class of the unknown classes, we randomly select $r \times 25,000$ samples from the $C_i$-th class of the external MNIST dataset and put them into $D_U^U$.

#### C.2.5  REGARDING THE LABEL DISTRIBUTION IN UNSEEN CLASSES

We determine the number of samples to be adopted from each unknown class to obtain $D_U^U$ based on $C_{ib}$.

Table 7: A detailed description for each metric.

| Metric | Range | Description | Interpretation |
|---|---|---|---|
| $R_{slope}$ | $(-\infty, \infty)$ | Indicates the overall trend of accuracy ($Acc(r)$) as unseen classes are added. Positive values imply increasing accuracy, while negative values suggest a decline. | A larger $R_{slope}$ (closer to 0 or positive) indicates better global robustness to unseen classes. |
| GM | $[0, \infty)$ | Measures the cumulative magnitude of accuracy deviations from the average accuracy ($A\bar{C}C$). Higher values imply larger fluctuations. | A smaller GM indicates higher stability and robustness to the addition of unseen classes. |
| WAD | $(-\infty, \infty)$ | Captures the largest negative accuracy drop between two adjacent $r$ values. | A larger WAD indicates better local robustness to unseen-class variations. |
| BAD | $(-\infty, \infty)$ | Reflects the maximum positive or minimal negative accuracy change between adjacent $r$ values. Positive values indicate improvements. | A larger BAD (positive or close to 0) indicates better performance in the best-case scenario. |
| $P_{AD\geq0}$ | $[0, 1]$ | Represents the probability that adding unseen classes does not degrade performance ($AD \geq 0$). | A higher $P_{AD\geq0}$ (close to 1) indicates better global robustness to unseen-class additions. |

Table 8: Evaluation on CIFAR100 with 1000 labels under the different number of unseen classes.

| METHOD | base | $C_n$=10 | $C_n$=20 | $C_n$=30 | $C_n$=40 | $C_n$=50 | $R_{slope} \times 10$ | GM | BAD×10 | WAD×10 | $P_{AD\geq0}$ |
|---|---|---|---|---|---|---|---|---|---|---|---|
| PSEUDOLABEL | 0.421 | 0.415 | 0.416 | 0.416 | 0.417 | 0.417 | 0.000 | 0.003 | 0.001 | 0.000 | 1.000 |
| PIMODEL | 0.424 | 0.428 | 0.426 | 0.423 | 0.425 | 0.422 | -0.001 | 0.009 | 0.002 | -0.003 | 0.250 |
| FIXMATCH | 0.441 | 0.384 | 0.338 | 0.398 | 0.394 | 0.332 | -0.005 | 0.137 | 0.060 | -0.062 | 0.250 |
| FLEXMATCH | 0.443 | 0.425 | 0.423 | 0.415 | 0.400 | 0.412 | -0.005 | 0.036 | 0.012 | -0.015 | 0.250 |
| UDA | 0.380 | 0.385 | 0.382 | 0.379 | 0.386 | 0.383 | 0.000 | 0.010 | 0.007 | -0.003 | 0.250 |
| SOFTMATCH | 0.455 | 0.447 | 0.440 | 0.441 | 0.452 | 0.449 | 0.002 | 0.021 | 0.011 | -0.007 | 0.500 |
| VAT | 0.446 | 0.426 | 0.430 | 0.426 | 0.435 | 0.435 | 0.002 | 0.018 | 0.009 | -0.004 | 0.750 |
| FREEMATCH | 0.455 | 0.431 | 0.421 | 0.426 | 0.439 | 0.435 | 0.003 | 0.028 | 0.013 | -0.010 | 0.500 |
| ICT | 0.413 | 0.415 | 0.413 | 0.411 | 0.413 | 0.414 | 0.000 | 0.005 | 0.002 | -0.002 | 0.500 |
| UASD | 0.415 | 0.409 | 0.415 | 0.414 | 0.413 | 0.411 | 0.000 | 0.010 | 0.006 | -0.002 | 0.250 |
| MTCF | 0.248 | 0.258 | 0.250 | 0.238 | 0.227 | 0.226 | -0.009 | 0.057 | -0.001 | -0.012 | 0.000 |
| CAFA | 0.424 | 0.421 | 0.413 | 0.417 | 0.421 | 0.420 | 0.001 | 0.014 | 0.004 | -0.008 | 0.500 |
| OPENMATCH | 0.367 | 0.352 | 0.327 | 0.347 | 0.346 | 0.329 | -0.003 | 0.049 | 0.020 | -0.025 | 0.250 |
| FIX_A_STEP | 0.448 | 0.413 | 0.400 | 0.395 | 0.408 | 0.393 | -0.003 | 0.035 | 0.013 | -0.015 | 0.250 |

## C.3 CATEGORY-NUMBER FACTOR IN CIFAR100

We validated different approaches on CIFAR100, with the results shown in Table 8. Here, $base$ refers to the scenario where $r$=0, meaning no unseen class data is added. In other cases, $r$ is set to 0.2. To ensure a sufficient sample size, $C_n$ is set to 10, 20, 30, 40, and 50, respectively. On the CIFAR100 dataset, the accuracy of all methods is generally lower, reflecting the complexity of the dataset. As $C_n$ increases from 10 to 50, the performance of many methods declines, but the rate of decline is more gradual compared to CIFAR10. The performance of MTCF decreases significantly, while PiModel and PseudoLabel change little, indicating that these methods are more robust to different Category-number factors. Overall, looking at the results of the two datasets, some methods show strong robustness and adaptability when dealing with different unseen class categories (such as ICT, CAFA).

## C.4 INCORPORATING NEARNESS FACTOR INTO CATEGORY-NUMBER FACTOR

We incorporate nearness factor into category-number factor and the experimental results can be seen in Table 9. By comparing Table 9 and Table 3 in main body, we can conclude that in the far OOD scenario, SSL models are more robust to changes in the number of unseen classes. This indicates that the farther the OOD, the smaller the impact on SSL models (whether positive or negative).

## C.5 THE IMPACT OF SEEN-CLASS RATIO $r_s$ FOR SSL MODEL

Previous works, such as DS3L Guo et al. (2020), when studying the impact of unseen classes on SSL models, employed a flawed approach that violated the principle of controlling variables. In their setup, they fixed the total number of unlabeled samples and merely adjusted the proportion of unseen class samples within this total. The decrease in SSL model performance, caused by the increase in the proportion of unseen classes, is not solely due to the increase in unseen class samples but

Table 9: Evaluation on CIFAR10 with 100 labels under the different number of unseen classes from far OOD scenario.

| METHOD | $base$ | $C_n$=1 | $C_n$=2 | $C_n$=3 | $C_n$=4 | $C_n$=5 | $R_{slope}$ | GM | BAD | WAD | $P_{AD \geq 0}$ |
|---|---|---|---|---|---|---|---|---|---|---|---|
| SUPERVISED | 0.617 | 0.617 | 0.617 | 0.617 | 0.617 | 0.617 | 0.000 | 0.000 | 0.000 | 0.000 | 1.000 |
| PSEUDOLABEL | 0.677 | 0.623 | 0.624 | 0.622 | 0.623 | 0.621 | 0.000 | 0.004 | 0.001 | -0.002 | 0.500 |
| PIMODEL | 0.667 | 0.624 | 0.624 | 0.625 | 0.624 | 0.622 | 0.000 | 0.004 | 0.001 | -0.002 | 0.500 |
| FIXMATCH | 0.612 | 0.702 | 0.703 | 0.678 | 0.689 | 0.691 | -0.004 | 0.040 | 0.011 | -0.025 | 0.750 |
| FLEXMATCH | 0.770 | 0.616 | 0.617 | 0.621 | 0.611 | 0.614 | -0.001 | 0.013 | 0.004 | -0.010 | 0.750 |
| UDA | 0.645 | 0.633 | 0.643 | 0.631 | 0.621 | 0.636 | -0.002 | 0.027 | 0.015 | -0.012 | 0.500 |
| SOFTMATCH | 0.764 | 0.642 | 0.656 | 0.639 | 0.642 | 0.644 | -0.001 | 0.023 | 0.014 | -0.017 | 0.750 |
| VAT | 0.710 | 0.581 | 0.578 | 0.588 | 0.585 | 0.582 | 0.001 | 0.015 | 0.010 | -0.003 | 0.250 |
| FREEMATCH | 0.760 | 0.637 | 0.644 | 0.631 | 0.628 | 0.635 | -0.002 | 0.022 | 0.007 | -0.013 | 0.500 |
| ICT | 0.618 | 0.597 | 0.598 | 0.599 | 0.598 | 0.599 | 0.000 | 0.003 | 0.001 | -0.001 | 0.750 |
| UASD | 0.618 | 0.617 | 0.619 | 0.618 | 0.619 | 0.618 | 0.000 | 0.003 | 0.002 | -0.001 | 0.500 |
| MTCF | 0.772 | 0.596 | 0.557 | 0.605 | 0.567 | 0.603 | 0.002 | 0.094 | 0.048 | -0.039 | 0.500 |
| CAFA | 0.652 | 0.621 | 0.620 | 0.621 | 0.617 | 0.608 | -0.003 | 0.020 | 0.001 | -0.009 | 0.250 |
| OPENMATCH | 0.713 | 0.622 | 0.617 | 0.614 | 0.605 | 0.598 | -0.006 | 0.039 | -0.003 | -0.009 | 0.000 |
| FIX_A_STEP | 0.622 | 0.693 | 0.682 | 0.689 | 0.677 | 0.690 | -0.001 | 0.027 | 0.013 | -0.012 | 0.500 |

Table 10: Evaluation on CIFAR10 with 100 labels when we fix $r_u$ and change $r_s$.

| METHOD | $r_s$=0 | $r_s$=0.2 | $r_s$=0.4 | $r_s$=0.5 | $r_s$=0.6 | $r_s$=0.8 | $r_s$=1.0 | $R_{slope}$ | GM | BAD | WAD | $P_{AD \geq 0}$ |
|---|---|---|---|---|---|---|---|---|---|---|---|---|
| SUPERVISED | 0.617 | 0.617 | 0.617 | 0.617 | 0.617 | 0.617 | 0.617 | 0.000 | 0.000 | 0.000 | 0.000 | 1.000 |
| PSEUDOLABEL | 0.653 | 0.669 | 0.672 | 0.673 | 0.675 | 0.674 | 0.675 | 0.018 | 0.037 | 0.080 | -0.005 | 0.833 |
| PIMODEL | 0.631 | 0.651 | 0.655 | 0.659 | 0.655 | 0.660 | 0.668 | 0.030 | 0.053 | 0.100 | -0.040 | 0.833 |
| FIXMATCH | 0.419 | 0.548 | 0.579 | 0.581 | 0.588 | 0.597 | 0.603 | 0.154 | 0.303 | 0.645 | 0.020 | 1.000 |
| FLEXMATCH | 0.626 | 0.718 | 0.749 | 0.747 | 0.757 | 0.752 | 0.756 | 0.109 | 0.229 | 0.460 | -0.025 | 0.667 |
| MIXMATCH | 0.678 | 0.708 | 0.722 | 0.723 | 0.728 | 0.731 | 0.729 | 0.047 | 0.096 | 0.150 | -0.010 | 0.833 |
| UDA | 0.543 | 0.621 | 0.640 | 0.648 | 0.638 | 0.641 | 0.654 | 0.088 | 0.178 | 0.390 | -0.100 | 0.833 |
| SOFTMATCH | 0.596 | 0.710 | 0.737 | 0.743 | 0.735 | 0.752 | 0.755 | 0.131 | 0.261 | 0.570 | -0.080 | 0.833 |
| VAT | 0.640 | 0.677 | 0.691 | 0.696 | 0.695 | 0.699 | 0.701 | 0.054 | 0.108 | 0.185 | -0.010 | 0.833 |
| FREEMATCH | 0.488 | 0.723 | 0.689 | 0.751 | 0.707 | 0.751 | 0.725 | 0.184 | 0.408 | 1.175 | -0.440 | 0.500 |
| ICT | 0.621 | 0.620 | 0.621 | 0.618 | 0.619 | 0.620 | 0.621 | 0.000 | 0.006 | 0.010 | -0.030 | 0.667 |
| UASD | 0.618 | 0.619 | 0.620 | 0.617 | 0.619 | 0.618 | 0.618 | -0.001 | 0.005 | 0.020 | -0.030 | 0.667 |
| MTCF | 0.689 | 0.744 | 0.759 | 0.762 | 0.762 | 0.766 | 0.772 | 0.069 | 0.136 | 0.275 | 0.000 | 1.000 |
| CAFA | 0.637 | 0.643 | 0.644 | 0.641 | 0.657 | 0.653 | 0.650 | 0.015 | 0.041 | 0.160 | -0.030 | 0.500 |
| OPENMATCH | 0.555 | 0.605 | 0.629 | 0.674 | 0.632 | 0.670 | 0.648 | 0.095 | 0.205 | 0.450 | -0.420 | 0.667 |
| FIX_A_STEP | 0.587 | 0.632 | 0.678 | 0.644 | 0.655 | 0.650 | 0.653 | 0.052 | 0.133 | 0.230 | -0.340 | 0.667 |

could also result from the decrease in seen class samples. To test this hypothesis, we conducted the following experiment: We fixed the number of unseen class samples in the unlabeled data and varied the number of seen class samples in the unlabeled data to verify the previous conjecture. The results are shown in Table 10. The table shows that when the ratio $r_s$ decreases from 1.0 to 0, almost all methods experience a significant performance drop, indicating that the quantity of seen class samples in the unlabeled data also has a significant impact on SSL models. This also proves that the previous evaluation strategy was flawed. That is, the previous evaluation strategy could not accurately describe the relationship between unseen classes and SSL models.

## C.6 EVALUATION ON A LARGER-SCALE DATASET

Based on the results in Table 11, we can see that under Imagenet-100, the influence of the addition of unseen classes on the SSL methods is smaller compared to CIFAR-10 and CIFAR-100. Most SSL methods exhibit similar properties across CIFAR-10, CIFAR-100, and Imagenet-100. For example, ICT and UASD demonstrate robustness to the addition of unseen classes across all three datasets. SoftMatch and FixMatch are relatively sensitive to the addition of unseen classes across all three datasets.

Additionally, similar to Tables 1 and 2, we also calculated five metrics to measure the degree to which semi-supervised learning models are affected by the addition of unseen classes on Imagenet-100, from both a global robustness and a local robustness perspective.

Table 11: Evaluation on ImageNet-100 with 5000 labels under inconsistent label spaces.

| METHOD | $r$=0 | $r$=0.2 | $r$=0.4 | $r$=0.5 | $r$=0.6 | $r$=0.8 | $r$=1.0 | $R_{slope}$ | GM | BAD | WAD | $P_{AD\geq0}$ | P-VALUE |
|---|---|---|---|---|---|---|---|---|---|---|---|---|---|
| PSEUDOLABEL | 0.258 | 0.253 | 0.262 | 0.257 | 0.261 | 0.263 | 0.258 | 0.004 | 0.018 | 0.045 | -0.050 | 0.500 | 0.541 |
| PIMODEL | 0.272 | 0.265 | 0.255 | 0.250 | 0.260 | 0.260 | 0.249 | -0.017 | 0.044 | 0.100 | -0.055 | 0.333 | 0.001 |
| FIXMATCH | 0.282 | 0.278 | 0.277 | 0.279 | 0.268 | 0.268 | 0.254 | -0.025 | 0.053 | 0.020 | -0.110 | 0.333 | 0.033 |
| FLEXMATCH | 0.291 | 0.291 | 0.291 | 0.287 | 0.284 | 0.281 | 0.285 | -0.009 | 0.023 | 0.019 | -0.040 | 0.500 | 0.039 |
| UDA | 0.240 | 0.246 | 0.248 | 0.256 | 0.255 | 0.252 | 0.256 | 0.014 | 0.034 | 0.080 | -0.015 | 0.666 | 0.001 |
| SOFTMATCH | 0.295 | 0.293 | 0.286 | 0.285 | 0.284 | 0.270 | 0.280 | -0.020 | 0.040 | 0.050 | -0.069 | 0.166 | 0.012 |
| VAT | 0.269 | 0.265 | 0.260 | 0.269 | 0.269 | 0.262 | 0.263 | -0.004 | 0.022 | 0.090 | -0.035 | 0.500 | 0.035 |
| FREEMATCH | 0.296 | 0.288 | 0.289 | 0.284 | 0.287 | 0.292 | 0.291 | -0.002 | 0.020 | 0.030 | -0.050 | 0.500 | 0.001 |
| ICT | 0.259 | 0.261 | 0.262 | 0.262 | 0.258 | 0.260 | 0.258 | -0.001 | 0.010 | 0.010 | -0.040 | 0.666 | 0.180 |
| UASD | 0.261 | 0.261 | 0.254 | 0.261 | 0.260 | 0.255 | 0.258 | -0.003 | 0.017 | 0.070 | -0.035 | 0.500 | 0.072 |
| MTCF | 0.094 | 0.094 | 0.096 | 0.110 | 0.099 | 0.102 | 0.098 | 0.006 | 0.027 | 0.136 | -0.106 | 0.666 | 0.028 |
| CAFA | 0.257 | 0.254 | 0.261 | 0.264 | 0.258 | 0.255 | 0.263 | 0.004 | 0.022 | 0.040 | -0.060 | 0.500 | 0.258 |
| OPENMATCH | 0.052 | 0.052 | 0.056 | 0.059 | 0.054 | 0.056 | 0.057 | 0.005 | 0.014 | 0.029 | -0.049 | 0.833 | 0.013 |
| FIX_A_STEP | 0.261 | 0.261 | 0.260 | 0.264 | 0.257 | 0.264 | 0.255 | -0.003 | 0.017 | 0.040 | -0.070 | 0.500 | 0.600 |

## C.7 EVALUATION ON A FINE-GRAINED DATASET

We conducted experiments for r=0, r=0.5, and r=1. The results are shown in the Table 12. Experimental observations are similar to CIFAR-10 and CIFAR-100, where the performance of most methods decreases as unseen classes are added (e.g., FreeMatch and FlexMatch), while some methods exhibit robustness to the addition of unseen classes (e.g., UASD and ICT).

Table 12: Evaluation on CUB with 5000 labels under inconsistent label spaces.

| METHOD | $r$=0 | $r$=0.5 | $r$=1.0 | $R_{slope}$ | P-VALUE |
|---|---|---|---|---|---|
| PI-MODEL | 0.487 | 0.487 | 0.507 | 0.001 | 0.422 |
| FIXMATCH | 0.500 | 0.504 | 0.494 | -0.005 | 0.859 |
| FLEXMATCH | 0.529 | 0.447 | 0.413 | -0.115 | 0.028 |
| SOFTMATCH | 0.510 | 0.517 | 0.477 | -0.033 | 0.582 |
| VAT | 0.510 | 0.490 | 0.490 | -0.019 | 0.000 |
| FREEMATCH | 0.509 | 0.448 | 0.410 | -0.098 | 0.052 |
| ICT | 0.473 | 0.473 | 0.473 | 0.000 | - |
| UASD | 0.668 | 0.668 | 0.668 | 0.000 | - |
| FIX-A-STEP | 0.394 | 0.391 | 0.393 | 0.000 | 0.183 |

## C.8 EVALUATION ON TABULAR DATASETS

Based on the results from Table 13 and Table 14, it can be observed that compared to image modality data, tabular data is less affected by unseen classes, and semi-supervised models demonstrate greater robustness on tabular data. We attribute this primarily to the simplicity of tabular data structure and the more effective feature representation. Moreover, similar to CIFAR, ImageNet, and CUB, ICT and UASD are relatively less affected by unseen classes compared to other semi-supervised methods, demonstrating stronger robustness against unseen classes.

Table 13: Evaluation on Forest under inconsistent label spaces.

| METHOD | $r=0$ | $r=0.2$ | $r=0.4$ | $r=0.6$ | $r=0.8$ | $r=1.0$ | $R_{slope}$ | GM | BAD | WAD | $P_{AD\geq0}$ | P-VALUE |
|---|---|---|---|---|---|---|---|---|---|---|---|---|
| PSEUDOLABEL | 0.610 | 0.603 | 0.606 | 0.607 | 0.604 | 0.603 | -0.004 | 0.011 | 0.010 | -0.032 | 0.400 | 0.000 |
| PIMODEL | 0.593 | 0.595 | 0.593 | 0.599 | 0.597 | 0.601 | 0.007 | 0.015 | 0.029 | -0.009 | 0.600 | 0.022 |
| FIXMATCH | 0.577 | 0.580 | 0.580 | 0.579 | 0.584 | 0.586 | 0.008 | 0.016 | 0.027 | -0.006 | 0.800 | 0.007 |
| FLEXMATCH | 0.584 | 0.591 | 0.591 | 0.590 | 0.586 | 0.593 | 0.004 | 0.016 | 0.036 | -0.018 | 0.600 | 0.001 |
| UDA | 0.609 | 0.608 | 0.606 | 0.606 | 0.609 | 0.604 | -0.003 | 0.010 | 0.015 | -0.023 | 0.200 | 0.024 |
| SOFTMATCH | 0.592 | 0.587 | 0.591 | 0.592 | 0.593 | 0.600 | 0.008 | 0.015 | 0.035 | -0.022 | 0.800 | 0.783 |
| VAT | 0.609 | 0.613 | 0.614 | 0.611 | 0.613 | 0.611 | 0.000 | 0.008 | 0.019 | -0.014 | 0.600 | 0.000 |
| FREEMATCH | 0.591 | 0.603 | 0.602 | 0.593 | 0.595 | 0.607 | 0.007 | 0.033 | 0.062 | -0.041 | 0.600 | 0.008 |
| ICT | 0.610 | 0.613 | 0.607 | 0.611 | 0.611 | 0.612 | 0.001 | 0.008 | 0.019 | -0.029 | 0.600 | 0.455 |
| UASD | 0.605 | 0.604 | 0.606 | 0.598 | 0.598 | 0.603 | -0.004 | 0.016 | 0.023 | -0.038 | 0.600 | 0.084 |
| MTCF | 0.590 | 0.592 | 0.594 | 0.596 | 0.596 | 0.598 | 0.007 | 0.013 | 0.010 | 0.001 | 1.000 | 0.001 |
| CAFA | 0.604 | 0.602 | 0.604 | 0.602 | 0.611 | 0.603 | 0.003 | 0.012 | 0.041 | -0.038 | 0.400 | 0.818 |
| OPENMATCH | 0.615 | 0.617 | 0.610 | 0.612 | 0.613 | 0.615 | -0.001 | 0.011 | 0.011 | -0.031 | 0.800 | 0.222 |
| FIX_A_STEP | 0.605 | 0.604 | 0.610 | 0.607 | 0.612 | 0.616 | 0.010 | 0.022 | 0.032 | -0.018 | 0.600 | 0.048 |

Table 14: Evaluation on Letter under inconsistent label spaces.

| METHOD | $r=0$ | $r=0.2$ | $r=0.4$ | $r=0.6$ | $r=0.8$ | $r=1.0$ | $R_{slope}$ | GM | BAD | WAD | $P_{AD\geq0}$ | P-VALUE |
|---|---|---|---|---|---|---|---|---|---|---|---|---|
| PSEUDOLABEL | 0.719 | 0.717 | 0.718 | 0.718 | 0.715 | 0.714 | -0.004 | 0.009 | 0.005 | -0.015 | 0.400 | 0.032 |
| PIMODEL | 0.726 | 0.719 | 0.716 | 0.715 | 0.717 | 0.710 | -0.012 | 0.021 | 0.008 | -0.037 | 0.200 | 0.002 |
| FIXMATCH | 0.740 | 0.718 | 0.708 | 0.689 | 0.679 | 0.688 | -0.056 | 0.110 | 0.044 | -0.110 | 0.200 | 0.003 |
| FLEXMATCH | 0.749 | 0.735 | 0.724 | 0.723 | 0.728 | 0.724 | -0.020 | 0.044 | 0.024 | -0.070 | 0.200 | 0.001 |
| UDA | 0.719 | 0.718 | 0.714 | 0.715 | 0.719 | 0.718 | 0.000 | 0.010 | 0.020 | -0.016 | 0.400 | 0.085 |
| SOFTMATCH | 0.726 | 0.712 | 0.707 | 0.711 | 0.717 | 0.715 | -0.005 | 0.027 | 0.026 | -0.072 | 0.400 | 0.001 |
| MEANTEACHER | 0.739 | 0.738 | 0.738 | 0.737 | 0.738 | 0.740 | 0.000 | 0.004 | 0.011 | -0.002 | 0.400 | 0.177 |
| VAT | 0.735 | 0.729 | 0.726 | 0.725 | 0.721 | 0.721 | -0.013 | 0.023 | 0.000 | -0.029 | 0.000 | 0.002 |
| FREEMATCH | 0.724 | 0.614 | 0.615 | 0.620 | 0.614 | 0.611 | -0.079 | 0.181 | 0.025 | -0.549 | 0.400 | 0.000 |
| ICT | 0.728 | 0.729 | 0.729 | 0.730 | 0.727 | 0.730 | 0.000 | 0.005 | 0.012 | -0.015 | 0.800 | 0.141 |
| UASD | 0.723 | 0.726 | 0.725 | 0.723 | 0.725 | 0.731 | 0.005 | 0.012 | 0.029 | -0.011 | 0.600 | 0.089 |
| MTCF | 0.706 | 0.607 | 0.544 | 0.542 | 0.529 | 0.512 | -0.172 | 0.332 | -0.011 | -0.495 | 0.000 | 0.001 |
| CAFA | 0.732 | 0.726 | 0.724 | 0.720 | 0.720 | 0.726 | -0.007 | 0.020 | 0.029 | -0.028 | 0.200 | 0.002 |
| OPENMATCH | 0.738 | 0.723 | 0.718 | 0.719 | 0.718 | 0.716 | -0.017 | 0.033 | 0.007 | -0.075 | 0.200 | 0.000 |
| FIX_A_STEP | 0.760 | 0.747 | 0.740 | 0.738 | 0.736 | 0.733 | -0.024 | 0.045 | -0.006 | -0.062 | 0.000 | 0.001 |

## C.9 EVALUATION ON TEXT DATASET

According to Table 15, compared to tabular data, text modality data is relatively more affected by unseen classes. We attribute this to the greater difficulty in feature representation and the higher complexity of text-based tasks. Consistent with previous experimental findings, UASD is relatively less affected by unseen classes compared to other semi-supervised methods, demonstrating stronger robustness against unseen classes.

Table 15: Evaluation on AGNews under inconsistent label spaces.

| METHOD | $r=0$ | $r=0.2$ | $r=0.4$ | $r=0.5$ | $r=0.6$ | $r=0.8$ | $r=1.0$ | $R_{slope}$ | GM | BAD | WAD | $P_{AD\geq0}$ | P-VALUE |
|---|---|---|---|---|---|---|---|---|---|---|---|---|---|
| PSEUDOLABEL | 0.960 | 0.865 | 0.917 | 0.850 | 0.893 | 0.916 | 0.868 | -0.047 | 0.212 | 0.430 | -0.670 | 0.500 | 0.001 |
| PIMODEL | 0.971 | 0.881 | 0.887 | 0.933 | 0.889 | 0.939 | 0.944 | 0.005 | 0.209 | 0.465 | -0.449 | 0.666 | 0.004 |
| FIXMATCH | 0.852 | 0.765 | 0.882 | 0.804 | 0.810 | 0.797 | 0.903 | 0.040 | 0.292 | 0.584 | -0.779 | 0.500 | 0.302 |
| FLEXMATCH | 0.734 | 0.823 | 0.957 | 0.803 | 0.891 | 0.874 | 0.867 | 0.108 | 0.380 | 0.881 | -1.543 | 0.500 | 0.001 |
| UDA | 0.858 | 0.781 | 0.845 | 0.889 | 0.927 | 0.902 | 0.890 | 0.086 | 0.253 | 0.442 | -0.384 | 0.500 | 0.529 |
| SOFTMATCH | 0.757 | 0.919 | 0.915 | 0.925 | 0.857 | 0.798 | 0.843 | 0.000 | 0.362 | 0.809 | -0.678 | 0.500 | 0.002 |
| MEANTEACHER | 0.964 | 0.942 | 0.967 | 0.941 | 0.955 | 0.941 | 0.941 | -0.018 | 0.071 | 0.142 | -0.260 | 0.500 | 0.015 |
| UASD | 0.960 | 0.970 | 0.919 | 0.959 | 0.958 | 0.955 | 0.959 | -0.002 | 0.070 | 0.400 | -0.255 | 0.500 | 0.395 |

## C.10 EXPANDED EXPERIMENTAL RESULTS WITH STANDARD DEVIATIONS

Standard deviations is important to provide a more comprehensive evaluation of the experimental results. We have added the standard deviations and the updated results can be seen in Table 16, Table 17, Table 18, and Table 19.

## C.11 EVALUATION ON OPEN-WORLD SEMI-SUPERVISED LEARNING METHODS

To expand the application scenarios of the RE-SSL framework, we also conducted validation in an open-world semi-supervised learning setting, exploring the impact of unseen classes in unlabeled data on the classification of seen classes and unseen classes by the semi-supervised model during the testing phase. Unlike safe semi-supervised settings, open-world semi-supervised learning also requires classification of unseen-class data during testing. We evaluated two representative open-world SSL methods, ORCA Kai-Di Cao & Leskovec (2022) and SORL Sun et al. (2024). The experimental results are shown in Table 20. According to the above results, we know the addition of unseen classes is beneficial for both seen-class classification and unseen-class classification in both the ORCA and SORL.

Table 16: Evaluation on CIFAR10 with 100 labels under inconsistent label spaces.

| METHOD | $r$=0 | $r$=0.2 | $r$=0.4 | $r$=0.5 | $r$=0.6 | $r$=0.8 | $r$=1.0 |
|---|---|---|---|---|---|---|---|
| PSEUDOLABEL | 0.677±0.010 | 0.668±0.010 | 0.664±0.011 | 0.660±0.011 | 0.660±0.010 | 0.660±0.011 | 0.654±0.013 |
| PIMODEL | 0.667±0.011 | 0.651±0.011 | 0.644±0.009 | 0.637±0.011 | 0.636±0.016 | 0.636±0.008 | 0.630±0.007 |
| FLEXMATCH | 0.770±0.016 | 0.717±0.017 | 0.702±0.017 | 0.704±0.006 | 0.704±0.009 | 0.686±0.015 | 0.638±0.005 |
| MIXMATCH | 0.731±0.013 | 0.708±0.019 | 0.708±0.017 | 0.706±0.015 | 0.701±0.017 | 0.697±0.016 | 0.676±0.019 |
| UDA | 0.645±0.025 | 0.621±0.022 | 0.604±0.030 | 0.601±0.029 | 0.591±0.023 | 0.590±0.024 | 0.552±0.023 |
| SOFTMATCH | 0.764±0.016 | 0.710±0.026 | 0.689±0.012 | 0.697±0.032 | 0.692±0.018 | 0.682±0.014 | 0.607±0.064 |
| VAT | 0.710±0.005 | 0.677±0.015 | 0.663±0.008 | 0.663±0.012 | 0.657±0.014 | 0.651±0.017 | 0.639±0.007 |
| FREEMATCH | 0.760±0.019 | 0.723±0.012 | 0.665±0.058 | 0.635±0.015 | 0.608±0.012 | 0.605±0.060 | 0.440±0.041 |
| ICT | 0.618±0.010 | 0.619±0.011 | 0.621±0.012 | 0.620±0.011 | 0.621±0.012 | 0.619±0.014 | 0.621±0.013 |
| UASD | 0.618±0.010 | 0.619±0.009 | 0.617±0.008 | 0.616±0.007 | 0.617±0.008 | 0.618±0.008 | 0.618±0.008 |
| MTCF | 0.772±0.014 | 0.743±0.009 | 0.731±0.017 | 0.723±0.010 | 0.725±0.015 | 0.716±0.015 | 0.692±0.015 |
| CAFA | 0.652±0.010 | 0.652±0.014 | 0.640±0.014 | 0.653±0.010 | 0.641±0.015 | 0.642±0.012 | 0.640±0.016 |
| OPENMATCH | 0.713±0.010 | 0.606±0.075 | 0.586±0.032 | 0.595±0.033 | 0.579±0.027 | 0.517±0.030 | 0.473±0.044 |
| FIX_A_STEP | 0.662±0.081 | 0.634±0.061 | 0.615±0.062 | 0.582±0.095 | 0.555±0.030 | 0.585±0.013 | 0.509±0.013 |

Table 17: Evaluation on CIFAR10 with 100 labels under the different number of unseen classes.

| METHOD | $C_n$=1 | $C_n$=2 | $C_n$=3 | $C_n$=4 | $C_n$=5 |
|---|---|---|---|---|---|
| PSEUDOLABEL | 0.648±0.008 | 0.652±0.012 | 0.663±0.009 | 0.664±0.011 | 0.668±0.010 |
| PIMODEL | 0.622±0.008 | 0.628±0.006 | 0.640±0.014 | 0.642±0.014 | 0.651±0.011 |
| FLEXMATCH | 0.658±0.034 | 0.644±0.015 | 0.689±0.008 | 0.699±0.009 | 0.715±0.019 |
| UDA | 0.620±0.017 | 0.599±0.010 | 0.594±0.035 | 0.621±0.014 | 0.621±0.022 |
| SOFTMATCH | 0.671±0.043 | 0.662±0.047 | 0.724±0.011 | 0.715±0.021 | 0.710±0.026 |
| VAT | 0.606±0.028 | 0.627±0.034 | 0.671±0.021 | 0.676±0.016 | 0.677±0.015 |
| FREEMATCH | 0.668±0.038 | 0.639±0.054 | 0.724±0.001 | 0.688±0.027 | 0.723±0.012 |
| ICT | 0.611±0.015 | 0.613±0.008 | 0.615±0.010 | 0.619±0.011 | 0.619±0.011 |
| UASD | 0.618±0.009 | 0.620±0.009 | 0.617±0.009 | 0.619±0.007 | 0.619±0.009 |
| MTCF | 0.723±0.007 | 0.704±0.016 | 0.737±0.009 | 0.736±0.012 | 0.743±0.009 |
| CAFA | 0.642±0.013 | 0.645±0.016 | 0.642±0.010 | 0.643±0.009 | 0.648±0.013 |
| OPENMATCH | 0.642±0.024 | 0.634±0.049 | 0.633±0.031 | 0.561±0.035 | 0.606±0.075 |
| FIX_A_STEP | 0.674±0.026 | 0.647±0.037 | 0.631±0.030 | 0.625±0.055 | 0.632±0.061 |

Table 18: Evaluation on CIFAR10 with 100 labels under the different index of unseen classes.

| METHOD | near | | | | | far | | | | |
|---|---|---|---|---|---|---|---|---|---|---|
| | $C_i$=5 | $C_i$=6 | $C_i$=7 | $C_i$=8 | $C_i$=9 | $C_i$=5 | $C_i$=6 | $C_i$=7 | $C_i$=8 | $C_i$=9 |
| PSEUDOLABEL | 0.648±0.014 | 0.649±0.015 | 0.650±0.015 | 0.650±0.016 | 0.648±0.008 | 0.614±0.015 | 0.618±0.019 | 0.626±0.012 | 0.621±0.012 | 0.623±0.013 |
| PIMODEL | 0.638±0.009 | 0.642±0.007 | 0.639±0.008 | 0.637±0.001 | 0.622±0.008 | 0.621±0.010 | 0.624±0.007 | 0.625±0.008 | 0.621±0.007 | 0.624±0.005 |
| FLEXMATCH | 0.656±0.027 | 0.640±0.015 | 0.682±0.013 | 0.662±0.010 | 0.658±0.008 | 0.610±0.059 | 0.586±0.055 | 0.619±0.042 | 0.606±0.049 | 0.616±0.043 |
| UDA | 0.627±0.041 | 0.646±0.019 | 0.627±0.037 | 0.608±0.017 | 0.620±0.017 | 0.644±0.017 | 0.640±0.021 | 0.640±0.018 | 0.643±0.016 | 0.633±0.021 |
| SOFTMATCH | 0.654±0.030 | 0.665±0.067 | 0.725±0.006 | 0.687±0.013 | 0.671±0.043 | 0.646±0.027 | 0.650±0.022 | 0.639±0.031 | 0.644±0.032 | 0.642±0.033 |
| VAT | 0.654±0.003 | 0.632±0.016 | 0.657±0.021 | 0.623±0.026 | 0.606±0.028 | 0.564±0.007 | 0.565±0.012 | 0.577±0.018 | 0.568±0.007 | 0.581±0.008 |
| FREEMATCH | 0.655±0.068 | 0.610±0.021 | 0.709±0.024 | 0.628±0.075 | 0.668±0.038 | 0.642±0.068 | 0.633±0.061 | 0.641±0.050 | 0.639±0.065 | 0.637±0.063 |
| ICT | 0.623±0.009 | 0.619±0.011 | 0.622±0.009 | 0.608±0.005 | 0.611±0.015 | 0.597±0.004 | 0.596±0.007 | 0.598±0.007 | 0.597±0.006 | 0.597±0.007 |
| UASD | 0.617±0.010 | 0.616±0.088 | 0.618±0.007 | 0.619±0.007 | 0.618±0.009 | 0.617±0.006 | 0.618±0.009 | 0.618±0.009 | 0.618±0.009 | 0.617±0.007 |
| MTCF | 0.703±0.006 | 0.685±0.026 | 0.697±0.014 | 0.689±0.026 | 0.723±0.007 | 0.632±0.014 | 0.573±0.013 | 0.619±0.010 | 0.596±0.056 | 0.596±0.025 |
| CAFA | 0.639±0.003 | 0.639±0.005 | 0.638±0.013 | 0.634±0.016 | 0.630±0.019 | 0.606±0.021 | 0.597±0.038 | 0.592±0.025 | 0.612±0.017 | 0.613±0.008 |
| OPENMATCH | 0.636±0.023 | 0.546±0.038 | 0.648±0.015 | 0.566±0.027 | 0.642±0.024 | 0.607±0.047 | 0.559±0.006 | 0.569±0.046 | 0.629±0.046 | 0.622±0.050 |
| FIX_A_STEP | 0.630±0.053 | 0.630±0.053 | 0.638±0.035 | 0.634±0.038 | 0.630±0.026 | 0.693±0.022 | 0.684±0.019 | 0.693±0.025 | 0.691±0.016 | 0.693±0.022 |

Table 19: Evaluation on CIFAR10 with 100 labels under different imbalance factors of unseen classes.

| METHOD | $C_{ib}$=0.01 | $C_{ib}$=0.02 | $C_{ib}$=0.05 | $C_{ib}$=0.10 | $C_{ib}$=0.20 |
|---|---|---|---|---|---|
| PSEUDOLABEL | 0.660±0.014 | 0.660±0.017 | 0.663±0.014 | 0.658±0.015 | 0.662±0.013 |
| PIMODEL | 0.647±0.012 | 0.649±0.008 | 0.642±0.010 | 0.646±0.008 | 0.642±0.013 |
| FLEXMATCH | 0.710±0.019 | 0.703±0.021 | 0.714±0.010 | 0.706±0.017 | 0.717±0.014 |
| UDA | 0.620±0.035 | 0.647±0.019 | 0.634±0.029 | 0.615±0.043 | 0.619±0.032 |
| SOFTMATCH | 0.730±0.007 | 0.731±0.014 | 0.725±0.009 | 0.721±0.011 | 0.698±0.033 |
| VAT | 0.681±0.016 | 0.682±0.013 | 0.682±0.012 | 0.678±0.012 | 0.675±0.019 |
| FREEMATCH | 0.730±0.013 | 0.723±0.016 | 0.674±0.067 | 0.670±0.063 | 0.677±0.029 |
| ICT | 0.622±0.010 | 0.621±0.011 | 0.621±0.009 | 0.622±0.009 | 0.622±0.008 |
| UASD | 0.618±0.008 | 0.618±0.008 | 0.617±0.008 | 0.618±0.008 | 0.617±0.009 |
| MTCF | 0.722±0.014 | 0.720±0.011 | 0.722±0.016 | 0.721±0.011 | 0.722±0.013 |
| CAFA | 0.647±0.008 | 0.650±0.008 | 0.643±0.011 | 0.648±0.013 | 0.646±0.014 |
| OPENMATCH | 0.613±0.068 | 0.613±0.074 | 0.620±0.077 | 0.650±0.006 | 0.639±0.007 |
| FIX_A_STEP | 0.685±0.038 | 0.638±0.066 | 0.632±0.060 | 0.633±0.064 | 0.628±0.066 |

Table 20: Evaluation of open-world semi-supervised learning methods on CIFAR-10 with 100 labels under inconsistent label spaces.

| METHOD | $r$=0.2 SEEN/UNSEEN | $r$=0.4 SEEN/UNSEEN | $r$=0.6 SEEN/UNSEEN | $r$=0.8 SEEN/UNSEEN | $r$=1.0 SEEN/UNSEEN |
|---|---|---|---|---|---|
| ORCA | 0.512/0.631 | 0.607/0.633 | 0.616/0.622 | 0.669/0.774 | 0.682/0.750 |
| SORL | 0.502/0.709 | 0.615/0.888 | 0.651/0.881 | 0.663/0.893 | 0.729/0.888 |

