# OpenReview forum: "Re-Evaluating the Impact of Unseen-Class Unlabeled Data on Semi-Supervised Learning Model"
_ICLR.cc/2025/Conference — ICLR 2025 Poster_

### Official Review · Reviewer_wdK8 · 2024-10-19

**Soundness:** 3
**Presentation:** 3
**Contribution:** 3
**Rating:** 6
**Confidence:** 3

**Summary:**

This paper proposes an evaluation framework for safe semi-supervised learning (SSL) with unseen class data. The paper observes that previous works fix the total number of seen and unseen class data, which may not reflect the true influence of unseen class data. Therefore, they fix the number of seen class data and change a number of quantities of unseen class data to investigate their influence. Extensive experiments are performed with much analysis.

**Strengths:**

- The shortcomings of previous evaluation approaches are new to the SSL field.
- Many experiments are performed, providing a solid and comprehensive evaluation for robust SSL.
- The experimental analysis is also very comprehensive.
- The paper is well written and easy to understand.

**Weaknesses:**

- My main concern is the practicality of the framework. The main purpose of the paper is to test the influence of *increasing numbers* of unseen class data on the classification performance on seen class data. The proposed metrics are mainly *one-order statistics* of the classification performance, which show the change in performance with more unseen class data. However, the main interest of SSL with unseen class data in real-world applications may not be focused on one-order statistics. In real-world applications, we only care about the performance of a certain number of unseen class data instead of the performance change with more data, although many papers provide the figures with different numbers of unseen class data. Therefore, although the framework is valid for different sizes of unseen class data, the practicability for real applications may be limited.
- Only CIFAR datasets were used for experiments. Since previous SSL benchmarks provide more datasets from different modalities, such as text or tabular data, it is beneficial to consider more text and tabular datasets.
- The standard deviations of the experimental results should also be included.
- Some typos should be checked. For example, in Eq. (1), it should be argmin. In Line 431, it should be Table 5.

**Questions:**

- Are the experimental results in tables the average accuracies with multiple seeds?
- Is $C_{ib}$ the ratio between unseen and seen data?
- Is each entry in Table 6 the mean of several $C$, i.e. the mean of each row?

---

> ### Author Response · Authors · 2024-11-24
> **Official Response to Reviewer wdK8 (1)**
>
> **Q1**: My main concern is the practicality of the framework. The main purpose of the paper is to test the influence of increasing numbers of unseen class data on the classification performance on seen class data. The proposed metrics are mainly one-order statistics of the classification performance, which show the change in performance with more unseen class data. However, the main interest of SSL with unseen class data in real-world applications may not be focused on one-order statistics. In real-world applications, we only care about the performance of a certain number of unseen class data instead of the performance change with more data, although many papers provide the figures with different numbers of unseen class data. Therefore, although the framework is valid for different sizes of unseen class data, the practicability for real applications may be limited.
>
>
> **A1**: Thank you for your insightful comments, and I completely understand your concern regarding the practicality of the framework. I agree that in real-world applications, the primary focus is often not on the change in performance as more unseen class data is introduced, but rather on the performance with a specific number of unseen class data.
>
> Indeed, fixing the size of the unlabeled set is a common approach in practical applications. In many real-world scenarios, we collect unlabeled data without knowing the exact proportion of unseen classes (i.e., the value of $r$. In such cases, the previous framework is indeed valuable.
>
> However, we also believe that evaluating semi-supervised models under a single $r$ value is not sufficient for understanding the robustness of the models. Our motivation for evaluating the impact of unseen classes in unlabeled data across different values of $r$ is to provide a more comprehensive understanding of how these factors influence model performance under varied conditions. We aim to offer a fair and standard evaluation across multiple scenarios, which is important for understanding the broader applicability of semi-supervised learning models.
>
> In the revised version, we have clarified that while both frameworks are valuable, they cater to different use cases: one for real-world applications where the size of the unlabeled set is fixed and another for assessing the general robustness of models under varying proportions of unseen class data.

---

> ### Author Response · Authors · 2024-11-24
> **Official Response to Reviewer wdK8 (2)**
>
> **Q2**: Only CIFAR datasets were used for experiments. Since previous SSL benchmarks provide more datasets from different modalities, such as text or tabular data, it is beneficial to consider more text and tabular datasets.
>
>
> **A2**: In response to your concern, we would like to clarify that, in addition to CIFAR-10 and CIFAR-100, we have conducted experiments on a broader range of datasets to enhance the credibility and generalizability of our results. These include:
>
>
> 1. **Image dataset**:
>    - **ImageNet-100**: A widely-used benchmark dataset providing a large-scale and realistic evaluation, used to analyze the influence of unseen class data on semi-supervised models.
>
> | $r$         | 0   | 0.2 | 0.4 | 0.5 | 0.6 | 0.8 | 1.0  | $R_{slope}$ | GM  | BAD | WAD | $P_{AD\ge 0}$ |
> |----------------|--------|-------|-------|-------|-------|-------|-------|--------|-------|-------|-------|------|
> | Fix-A-Step     | 0.261  | 0.261 | 0.260 | 0.264 | 0.257 | 0.264 | 0.255 | -0.003 | 0.017 | 0.040 | -0.070| 0.500|
> | FixMatch       | 0.282  | 0.278 | 0.277 | 0.279 | 0.268 | 0.268 | 0.254 | -0.025 | 0.053 | 0.020 | -0.110| 0.333|
> | FlexMatch      | 0.291  | 0.291 | 0.291 | 0.287 | 0.284 | 0.281 | 0.285 | -0.009 | 0.023 | 0.019 | -0.040| 0.500|
> | FreeMatch      | 0.296  | 0.288 | 0.289 | 0.284 | 0.287 | 0.292 | 0.291 | -0.002 | 0.020 | 0.030 | -0.050| 0.500|
> | MTCF           | 0.094  | 0.094 | 0.096 | 0.096 | 0.099 | 0.102 | 0.098 |  0.006 | 0.027 | 0.136 | -0.106| 0.666|
> | OpenMatch      | 0.052  | 0.052 | 0.056 | 0.059 | 0.054 | 0.056 | 0.057 |  0.005 | 0.014 | 0.029 | -0.049| 0.833|
> | UASD           | 0.261  | 0.261 | 0.254 | 0.261 | 0.260 | 0.255 | 0.258 | -0.003 | 0.017 | 0.070 | -0.035| 0.500|
> | ICT            | 0.259  | 0.261 | 0.262 | 0.262 | 0.258 | 0.260 | 0.258 | -0.001 | 0.010 | 0.010 | -0.040| 0.666|
> | PI-Model       | 0.272  | 0.265 | 0.255 | 0.250 | 0.260 | 0.260 | 0.249 | -0.017 | 0.044 | 0.100 | -0.055| 0.333|
> | Pseudo-Label   | 0.258  | 0.253 | 0.262 | 0.257 | 0.261 | 0.263 | 0.258 |  0.004 | 0.018 | 0.045 | -0.050| 0.500 |
> | CAFA           | 0.257  | 0.254 | 0.261 | 0.264 | 0.258 | 0.255 | 0.263 |  0.004 | 0.022 | 0.040 | -0.060| 0.500|
> | SoftMatch      | 0.295  | 0.293 | 0.286 | 0.285 | 0.284 | 0.270 | 0.280 | -0.020 | 0.040 | 0.050 | -0.069| 0.166|
> | UDA            | 0.240  | 0.246 | 0.248 | 0.256 | 0.255 | 0.252 | 0.256 |  0.014 | 0.034 | 0.080 | -0.015| 0.666|
> | VAT            | 0.269  | 0.265 | 0.260 | 0.269 | 0.269 | 0.262 | 0.263 | -0.004 | 0.022 | 0.090 | -0.035| 0.500|
>
> Based on the results above, we can see that under Imagenet-100, the influence of the addition of unseen classes on the SSL methods is smaller compared to CIFAR-10 and CIFAR-100. Most SSL methods exhibit similar properties across CIFAR-10, CIFAR-100, and Imagenet-100. For example, ICT and UASD demonstrate robustness to the addition of unseen classes across all three datasets. SoftMatch and FixMatch are relatively sensitive to the addition of unseen classes across all three datasets.
>
> Additionally, similar to Tables 1 and 2 in the original submission, we also calculated five metrics to measure the degree to which semi-supervised learning models are affected by the addition of unseen classes on Imagenet-100, from both a global robustness and a local robustness perspective.
>
>    - **CUB**: A fine-grained classification dataset that introduces additional diversity, enabling the study of the effects of unseen class data in specific scenarios.
>
> | $r$         |   0    |  0.5  | 1.0   | $R_{slope}$  |
> |-------------|--------|-------|-------|---------|
> | PI-Model    | 0.487  | 0.487 | 0.507 | 0.001   |
> | UASD        | 0.668  | 0.668 | 0.668 | 0.000   |
> | FixMatch    | 0.500  | 0.504 | 0.494 | -0.005  |
> | FlexMatch   | 0.529  | 0.447 | 0.413 | -0.115  |
> | Fix-A-Step  | 0.394  | 0.391 | 0.393 | 0.000   |
> | SoftMatch   | 0.510  | 0.517 | 0.477 | -0.033  |
> | VAT         | 0.510  | 0.490 | 0.490 | -0.019  |
> | FreeMatch   | 0.509  | 0.448 | 0.410 | -0.098  |
> | ICT         | 0.473  | 0.473 | 0.473 | 0.000   |
>
> Due to time and resource constraints, we only conducted experiments for r=0, r=0.5, and r=1. The results are shown in the table above. Experimental observations are similar to CIFAR10 and CIFAR100, where the performance of most methods decreases as unseen classes are added (e.g., FreeMatch and FlexMatch), while some methods exhibit robustness to the addition of unseen classes (e.g., UASD and ICT).

---

> ### Author Response · Authors · 2024-11-24
> **Official Response to Reviewer wdK8 (3)**
>
> 2. **Tabular dataset**:
>    - **Forest**: A structured tabular dataset, used to investigate the impact of unseen class data in non-image modalities.
>
> | $r$               | 0 | 0.2 | 0.4 | 0.6 | 0.8 | 1.0 | $R_{slope}$ | GM  | BAD | WAD | $P_{AD\ge 0}$|
> |---------------------|-----------|-----------|-----------|-----------|-----------|-----------|--------|-------|-------|-------|-------|
> | MeanTeacher         | 0.619     | 0.617     | 0.618     | 0.619     | 0.620     | 0.619     | 0.001  | 0.004 | 0.006 | -0.009 | 0.600 |
> | FreeMatch           | 0.591     | 0.603     | 0.602     | 0.593     | 0.595     | 0.607     | 0.007  | 0.033 | 0.062 | -0.041 | 0.600 |
> | VAT                 | 0.609     | 0.613     | 0.614     | 0.611     | 0.613     | 0.611     | 0.000  | 0.008 | 0.019 | -0.014 | 0.600 |
> | UASD                | 0.605     | 0.604     | 0.606     | 0.598     | 0.598     | 0.603     | -0.004 | 0.016 | 0.023 | -0.038 | 0.600 |
> | MTCF                | 0.590     | 0.592     | 0.594     | 0.596     | 0.596     | 0.598     | 0.007  | 0.013 | 0.010 | 0.001  | 1.000 |
> | OpenMatch           | 0.615     | 0.617     | 0.610     | 0.612     | 0.613     | 0.615     | -0.001 | 0.011 | 0.011 | -0.031 | 0.800 |
> | Fix_A_Step          | 0.605     | 0.604     | 0.610     | 0.607     | 0.612     | 0.616     | 0.010  | 0.022 | 0.032 | -0.018 | 0.600 |
> | CAFA                | 0.604     | 0.602     | 0.604     | 0.602     | 0.611     | 0.603     | 0.003  | 0.012 | 0.041 | -0.038 | 0.400 |
> | PiModel             | 0.593     | 0.595     | 0.593     | 0.599     | 0.597     | 0.601     | 0.007  | 0.015 | 0.029 | -0.009 | 0.600 |
> | PseudoLabel         | 0.610     | 0.603     | 0.606     | 0.607     | 0.604     | 0.603     | -0.004 | 0.011 | 0.010 | -0.032 | 0.400 |
> | ICT                 | 0.610     | 0.613     | 0.607     | 0.611     | 0.611     | 0.612     | 0.001  | 0.008 | 0.019 | -0.029 | 0.600 |
> | MixMatch            | 0.608     | 0.613     | 0.612     | 0.616     | 0.604     | 0.610     | -0.002 | 0.019 | 0.033 | -0.061 | 0.600 |
> | FixMatch            | 0.577     | 0.580     | 0.580     | 0.579     | 0.584     | 0.586     | 0.008  | 0.016 | 0.027 | -0.006 | 0.800 |
> | FlexMatch           | 0.584     | 0.591     | 0.591     | 0.590     | 0.586     | 0.593     | 0.004  | 0.016 | 0.036 | -0.018 | 0.600 |
> | SoftMatch           | 0.592     | 0.587     | 0.591     | 0.592     | 0.593     | 0.600     | 0.008  | 0.015 | 0.035 | -0.022 | 0.800 |

---

> ### Author Response · Authors · 2024-11-24
> **Official Response to Reviewer wdK8 (4)**
>
> - **Letter**: Another tabular dataset included to further explore the influence of unseen class data on semi-supervised models.
>
> | $r$               | 0 | 0.2 | 0.4 | 0.6 | 0.8 | 1.0 | $R_{slope}$ | GM  | BAD | WAD | $P_{AD\ge 0}$|
> |---------------------|-----------|-----------|-----------|-----------|-----------|-----------|--------|-------|-------|-------|-------|
> | CAFA                | 0.732     | 0.726     | 0.724     | 0.720     | 0.720     | 0.726     | -0.007 | 0.020 | 0.029 | -0.028 | 0.200 |
> | PiModel             | 0.726     | 0.719     | 0.716     | 0.715     | 0.717     | 0.710     | -0.012 | 0.021 | 0.008 | -0.037 | 0.200 |
> | PseudoLabel         | 0.719     | 0.717     | 0.718     | 0.718     | 0.715     | 0.714     | -0.004 | 0.009 | 0.005 | -0.015 | 0.400 |
> | ICT                 | 0.728     | 0.729     | 0.729     | 0.730     | 0.727     | 0.730     | 0.000  | 0.005 | 0.012 | -0.015 | 0.800 |
> | MixMatch            | 0.680     | 0.680     | 0.685     | 0.680     | 0.677     | 0.678     | -0.003 | 0.011 | 0.026 | -0.024 | 0.400 |
> | FixMatch            | 0.740     | 0.718     | 0.708     | 0.689     | 0.679     | 0.688     | -0.056 | 0.110 | 0.044 | -0.110 | 0.200 |
> | FlexMatch           | 0.749     | 0.735     | 0.724     | 0.723     | 0.728     | 0.724     | -0.020 | 0.044 | 0.024 | -0.070 | 0.200 |
> | SoftMatch           | 0.726     | 0.712     | 0.707     | 0.711     | 0.717     | 0.715     | -0.005 | 0.027 | 0.026 | -0.072 | 0.400 |
> | MeanTeacher         | 0.739     | 0.738     | 0.738     | 0.737     | 0.738     | 0.740     | 0.000  | 0.004 | 0.011 | -0.002 | 0.400 |
> | FreeMatch           | 0.724     | 0.614     | 0.615     | 0.620     | 0.614     | 0.611     | -0.079 | 0.181 | 0.025 | -0.549 | 0.400 |
> | VAT                 | 0.735     | 0.729     | 0.726     | 0.725     | 0.721     | 0.721     | -0.013 | 0.023 | 0.000 | -0.029 | 0.000 |
> | UASD                | 0.723     | 0.726     | 0.725     | 0.723     | 0.725     | 0.731     | 0.005  | 0.012 | 0.029 | -0.011 | 0.600 |
> | MTCF                | 0.706     | 0.607     | 0.544     | 0.542     | 0.529     | 0.512     | -0.172 | 0.332 | -0.011 | -0.495 | 0.000 |
> | OpenMatch           | 0.738     | 0.723     | 0.718     | 0.719     | 0.718     | 0.716     | -0.017 | 0.033 | 0.007 | -0.075 | 0.200 |
> | Fix_A_Step          | 0.760     | 0.747     | 0.740     | 0.738     | 0.736     | 0.733     | -0.024 | 0.045 | -0.006 | -0.062 | 0.000 |
>
> Based on the results from the two tables above, it can be observed that compared to image modality data, tabular data is less affected by unseen classes, and semi-supervised models demonstrate greater robustness on tabular data. We attribute this primarily to the simplicity of tabular data structure and the more effective feature representation. Moreover, similar to CIFAR, ImageNet, and CUB, ICT and UASD are relatively less affected by unseen classes compared to other semi-supervised methods, demonstrating stronger robustness against unseen classes.

---

> ### Author Response · Authors · 2024-11-24
> **Official Response to Reviewer wdK8 (5)**
>
> 3. **Text dataset**:
>    - **AGNews**: A widely-used text dataset in natural language processing, included to assess how unseen class data affects semi-supervised models in text-based tasks.
>
> | $r$               | 0 | 0.2 | 0.4 |0.5| 0.6 | 0.8 | 1.0 | $R_{slope}$ | GM  | BAD | WAD | $P_{AD\ge 0}$|
> |---------------|-----------|-----------|-----------|-----------|-----------|-----------|-----------|--------|-------|-------|-------|-------|
> | PiModel       | 0.971     | 0.881     | 0.887     | 0.933     | 0.889     | 0.939     | 0.944     | 0.005  | 0.209 | 0.465 | -0.449 | 0.666 |
> | UDA           | 0.858     | 0.781     | 0.845     | 0.889     | 0.927     | 0.902     | 0.890     | 0.086  | 0.253 | 0.442 | -0.384 | 0.500 |
> | PseudoLabel   | 0.960     | 0.865     | 0.917     | 0.850     | 0.893     | 0.916     | 0.868     | -0.047 | 0.212 | 0.430 | -0.670 | 0.500 |
> | FixMatch      | 0.852     | 0.765     | 0.882     | 0.804     | 0.810     | 0.797     | 0.903     | 0.040  | 0.292 | 0.584 | -0.779 | 0.500 |
> | FlexMatch     | 0.734     | 0.823     | 0.957     | 0.803     | 0.891     | 0.874     | 0.867     | 0.108  | 0.380 | 0.881 | -1.543 | 0.500 |
> | SoftMatch     | 0.757     | 0.919     | 0.915     | 0.925     | 0.857     | 0.798     | 0.843     | 0.000  | 0.362 | 0.809 | -0.678 | 0.500 |
> | MeanTeacher   | 0.964     | 0.942     | 0.967     | 0.941     | 0.955     | 0.941     | 0.941     | -0.018 | 0.071 | 0.142 | -0.260 | 0.500 |
> | UASD          | 0.960     | 0.970     | 0.919     | 0.959     | 0.958     | 0.955     | 0.959     | -0.002 | 0.070 | 0.400 | -0.255 | 0.500 |
>
> Compared to tabular data, text modality data is relatively more affected by unseen classes. We attribute this to the greater difficulty in feature representation and the higher complexity of text-based tasks. Consistent with previous experimental findings, UASD is relatively less affected by unseen classes compared to other semi-supervised methods, demonstrating stronger robustness against unseen classes.
>
> These additional datasets aim to validate the effectiveness of our experimental setup in various domains, showcasing its practicality. We have incorporated the above experimental results and analysis into the revised manuscript.

---

> ### Author Response · Authors · 2024-11-24
> **Official Response to Reviewer wdK8 (6)**
>
> **Q3**: The standard deviations of the experimental results should also be included.
>
>
> **A3**: Thank you for your suggestion. We agree that including standard deviations is important to provide a more comprehensive evaluation of the experimental results. We have added the standard deviations and the updated results are shown in the table below:
>
> | $r_u$      | 0         | 0.2      | 0.4      | 0.5       | 0.6      | 0.8       | 1.0       |
> |------------------|-----------------|-----------------|-----------------|-----------------|-----------------|-----------------|-----------------|
> | PseudoLabel      | 0.677±0.010    | 0.668±0.010    | 0.664±0.011    | 0.660±0.011    | 0.660±0.010    | 0.660±0.011    | 0.654±0.013    |
> | PiModel          | 0.667±0.011    | 0.651±0.011    | 0.644±0.009    | 0.637±0.011    | 0.636±0.016    | 0.636±0.008    | 0.630±0.007    |
> | FlexMatch        | 0.770±0.016    | 0.717±0.017    | 0.702±0.017    | 0.704±0.006    | 0.704±0.009    | 0.686±0.015    | 0.638±0.005    |
> | MixMatch         | 0.731±0.013    | 0.708±0.019    | 0.708±0.017    | 0.706±0.015    | 0.701±0.017    | 0.697±0.016    | 0.676±0.019    |
> | UDA              | 0.645±0.025    | 0.621±0.022    | 0.604±0.030    | 0.601±0.029    | 0.591±0.023    | 0.590±0.024    | 0.552±0.023    |
> | SoftMatch        | 0.764±0.016    | 0.710±0.026    | 0.689±0.012    | 0.697±0.032    | 0.692±0.018    | 0.682±0.014    | 0.607±0.064    |
> | VAT              | 0.710±0.005    | 0.677±0.015    | 0.663±0.008    | 0.663±0.012    | 0.657±0.014    | 0.651±0.017    | 0.639±0.007    |
> | FreeMatch        | 0.760±0.019    | 0.723±0.012    | 0.665±0.058    | 0.635±0.015    | 0.608±0.012    | 0.605±0.060    | 0.440±0.041    |
> | ICT              | 0.618±0.010    | 0.619±0.011    | 0.621±0.012    | 0.620±0.011    | 0.621±0.012    | 0.619±0.014    | 0.621±0.013    |
> | UASD         | 0.618±0.010    | 0.619±0.009    | 0.617±0.008    | 0.616±0.007    | 0.617±0.008    | 0.618±0.008    | 0.618±0.008    |
> | MTCF         | 0.772±0.012    | 0.743±0.009    | 0.731±0.017    | 0.723±0.010    | 0.725±0.015    | 0.716±0.015    | 0.692±0.015    |
> | CAFA         | 0.652±0.010    | 0.652±0.014    | 0.640±0.014    | 0.653±0.010    | 0.641±0.015    | 0.642±0.012    | 0.640±0.016    |
> | OpenMatch    | 0.713±0.010    | 0.606±0.075    | 0.586±0.032    | 0.595±0.033    | 0.579±0.027    | 0.517±0.030    | 0.473±0.044    |
> | Fix\_A\_Step | 0.662±0.081    | 0.634±0.061    | 0.615±0.062    | 0.582±0.095    | 0.555±0.030    | 0.585±0.013    | 0.509±0.013    |
>
>
> | $C_n$ |      1      |      2      |      3       |      4       |      5       |
> |------------------|---------------|---------------|---------------|---------------|---------------|
> | PseudoLabel      | 0.648±0.008   | 0.652±0.012   | 0.663±0.009   | 0.664±0.011   | 0.668±0.010   |
> | PiModel          | 0.622±0.008   | 0.628±0.006   | 0.640±0.014   | 0.642±0.014   | 0.651±0.011   |
> | FlexMatch        | 0.658±0.034   | 0.644±0.015   | 0.689±0.008   | 0.699±0.009   | 0.715±0.019   |
> | UDA              | 0.620±0.017   | 0.599±0.010   | 0.594±0.035   | 0.621±0.014   | 0.621±0.022   |
> | SoftMatch        | 0.671±0.043   | 0.662±0.047   | 0.724±0.011   | 0.715±0.021   | 0.710±0.026   |
> | VAT              | 0.606±0.028   | 0.627±0.034   | 0.671±0.021   | 0.676±0.016   | 0.677±0.015   |
> | FreeMatch        | 0.668±0.038   | 0.639±0.054   | 0.724±0.001   | 0.688±0.027   | 0.723±0.012   |
> | ICT              | 0.611±0.015   | 0.613±0.008   | 0.615±0.010   | 0.619±0.011   | 0.619±0.011   |
> | UASD         | 0.618±0.009   | 0.620±0.009   | 0.617±0.009   | 0.619±0.007   | 0.619±0.009   |
> | MTCF         | 0.723±0.007   | 0.704±0.016   | 0.737±0.009   | 0.736±0.012   | 0.743±0.009   |
> | CAFA         | 0.642±0.013   | 0.645±0.016   | 0.642±0.010   | 0.643±0.009   | 0.648±0.013   |
> | OpenMatch    | 0.642±0.024   | 0.634±0.049   | 0.633±0.031   | 0.561±0.035   | 0.606±0.075   |
> | Fix\_A\_Step | 0.674±0.026   | 0.647±0.037   | 0.631±0.030   | 0.625±0.055   | 0.632±0.061   |

---

> ### Author Response · Authors · 2024-11-24
> **Official Response to Reviewer wdK8 (7)**
>
> | Method      | $C_i$=5           | $C_i$=6           | $C_i$=7           | $C_i$=8           | $C_i$=9           | $C_i$=5           | $C_i$=6           | $C_i$=7           | $C_i$=8           | $C_i$=9           |
> |-------------|-------------------|-------------------|-------------------|-------------------|-------------------|-------------------|-------------------|-------------------|-------------------|-------------------|
> |             | **near**           | **near**           | **near**           | **near**           | **near**           | **far**            | **far**            | **far**            | **far**            | **far**            |
> | PseudoLabel | 0.648 ± 0.014      | 0.649 ± 0.015      | 0.650 ± 0.015      | 0.650 ± 0.016      | 0.648 ± 0.008      | 0.614 ± 0.015      | 0.618 ± 0.019      | 0.626 ± 0.012      | 0.621 ± 0.012      | 0.623 ± 0.013      |
> | PiModel     | 0.638 ± 0.009      | 0.642 ± 0.007      | 0.639 ± 0.008      | 0.637 ± 0.001      | 0.622 ± 0.008      | 0.621 ± 0.010      | 0.624 ± 0.007      | 0.625 ± 0.008      | 0.621 ± 0.007      | 0.624 ± 0.005      |
> | FlexMatch   | 0.656 ± 0.027      | 0.640 ± 0.015      | 0.682 ± 0.013      | 0.662 ± 0.010      | 0.658 ± 0.034      | 0.610 ± 0.059      | 0.586 ± 0.055      | 0.619 ± 0.042      | 0.606 ± 0.049      | 0.616 ± 0.043      |
> | UDA         | 0.627 ± 0.041      | 0.646 ± 0.019      | 0.627 ± 0.037      | 0.608 ± 0.017      | 0.620 ± 0.017      | 0.644 ± 0.017      | 0.640 ± 0.021      | 0.640 ± 0.018      | 0.643 ± 0.016      | 0.633 ± 0.021      |
> | SoftMatch   | 0.654 ± 0.030      | 0.665 ± 0.067      | 0.725 ± 0.006      | 0.687 ± 0.013      | 0.671 ± 0.043      | 0.646 ± 0.027      | 0.650 ± 0.022      | 0.639 ± 0.031      | 0.644 ± 0.032      | 0.642 ± 0.033      |
> | VAT         | 0.654 ± 0.003      | 0.632 ± 0.016      | 0.657 ± 0.021      | 0.623 ± 0.026      | 0.606 ± 0.028      | 0.564 ± 0.007      | 0.565 ± 0.012      | 0.577 ± 0.018      | 0.568 ± 0.007      | 0.581 ± 0.008      |
> | FreeMatch   | 0.655 ± 0.068      | 0.610 ± 0.021      | 0.709 ± 0.024      | 0.628 ± 0.075      | 0.668 ± 0.038      | 0.642 ± 0.068      | 0.633 ± 0.061      | 0.641 ± 0.050      | 0.639 ± 0.065      | 0.637 ± 0.063      |
> | ICT         | 0.623 ± 0.009      | 0.619 ± 0.011      | 0.622 ± 0.009      | 0.608 ± 0.005      | 0.611 ± 0.015      | 0.597 ± 0.004      | 0.596 ± 0.007      | 0.598 ± 0.007      | 0.597 ± 0.006      | 0.597 ± 0.007      |
> | UASD        | 0.617 ± 0.010      | 0.616 ± 0.088      | 0.618 ± 0.007      | 0.619 ± 0.007      | 0.618 ± 0.009      | 0.617 ± 0.006      | 0.618 ± 0.009      | 0.618 ± 0.009      | 0.618 ± 0.009      | 0.617 ± 0.007      |
> | MTCF        | 0.703 ± 0.006      | 0.685 ± 0.026      | 0.697 ± 0.014      | 0.689 ± 0.026      | 0.723 ± 0.007      | 0.632 ± 0.014      | 0.573 ± 0.013      | 0.619 ± 0.010      | 0.596 ± 0.056      | 0.596 ± 0.025      |
> | CAFA        | 0.639 ± 0.003      | 0.639 ± 0.005      | 0.638 ± 0.013      | 0.634 ± 0.016      | 0.630 ± 0.019      | 0.606 ± 0.021      | 0.597 ± 0.038      | 0.592 ± 0.025      | 0.612 ± 0.017      | 0.613 ± 0.008      |
> | OpenMatch   | 0.636 ± 0.023      | 0.546 ± 0.038      | 0.648 ± 0.015      | 0.566 ± 0.027      | 0.642 ± 0.024      | 0.607 ± 0.047      | 0.559 ± 0.006      | 0.569 ± 0.046      | 0.629 ± 0.046      | 0.622 ± 0.050      |
> | Fix_A_Step  | 0.630 ± 0.053      | 0.630 ± 0.053      | 0.638 ± 0.035      | 0.634 ± 0.038      | 0.630 ± 0.026      | 0.693 ± 0.022      | 0.684 ± 0.019      | 0.693 ± 0.025      | 0.691 ± 0.016      | 0.693 ± 0.022      |

---

> ### Author Response · Authors · 2024-11-24
> **Official Response to Reviewer wdK8 (8)**
>
> | $C_{ib}$  |    0.01     |    0.02     |    0.05     |    0.10     |     0.20     |
> |---------------|---------------|---------------|---------------|---------------|---------------|
> | PseudoLabel   | 0.660±0.014    | 0.660±0.017    | 0.663±0.014    | 0.658±0.015    | 0.662±0.013    |
> | PiModel       | 0.647±0.012    | 0.649±0.008    | 0.642±0.010    | 0.646±0.008    | 0.642±0.013    |
> | FlexMatch     | 0.710±0.019    | 0.703±0.021    | 0.714±0.010    | 0.706±0.017    | 0.717±0.014    |
> | UDA           | 0.620±0.035    | 0.647±0.019    | 0.634±0.029    | 0.615±0.043    | 0.619±0.032    |
> | SoftMatch     | 0.730±0.007    | 0.731±0.014    | 0.725±0.009    | 0.721±0.011    | 0.698±0.033    |
> | VAT           | 0.681±0.016    | 0.682±0.013    | 0.682±0.012    | 0.678±0.012    | 0.675±0.019    |
> | FreeMatch     | 0.730±0.013    | 0.723±0.016    | 0.674±0.067    | 0.670±0.063    | 0.677±0.029    |
> | ICT           | 0.622±0.010    | 0.621±0.011    | 0.621±0.009    | 0.622±0.009    | 0.622±0.008    |
> | UASD          | 0.618±0.008    | 0.618±0.008    | 0.617±0.008    | 0.618±0.008    | 0.617±0.009    |
> | MTCF          | 0.722±0.014    | 0.720±0.011    | 0.722±0.016    | 0.721±0.011    | 0.722±0.013    |
> | CAFA          | 0.647±0.008    | 0.650±0.008    | 0.643±0.011    | 0.648±0.013    | 0.646±0.014    |
> | OpenMatch     | 0.613±0.068    | 0.613±0.074    | 0.620±0.077    | 0.650±0.006    | 0.639±0.007    |
> | Fix_A_Step    | 0.685±0.038    | 0.638±0.066    | 0.632±0.060    | 0.633±0.064    | 0.628±0.066    |
>
> We have updated the experimental results mentioned above in the Appendix of the revised manuscript.

---

> ### Author Response · Authors · 2024-11-24
> **Official Response to Reviewer wdK8 (9)**
>
> **Q4**: Some typos should be checked. For example, in Eq. (1), it should be argmin. In Line 431, it should be Table 5.
>
> **A4**: Thank you for pointing out the typos. We have carefully reviewed the manuscript to correct these errors. Specifically, we have updated Eq.(1) to use "argmin" as suggested, and we have corrected the reference to "Table 5" in Line 431. We appreciate your attention to these details and have ensured that the manuscript is free from such typographical errors in the revised version.
>
> **Q5**: Are the experimental results in tables the average accuracies with multiple seeds?
>
> **A5**: Thank you for your question. Yes, the experimental results presented in the tables are the average accuracies obtained from multiple runs with different random seeds. Specifically, we conducted the experiments using seeds 0, 1, and 2, and the reported results are the average of these runs. This approach helps ensure the robustness and reliability of the results.
>
> **Q6**: Is $C_{ib}$ the ratio between unseen and seen data?
>
> **A6**: Thank you for your question. $C_{ib}$ is not the ratio between unseen and seen data. $C_{ib}$, or the imbalance factor, controls the distribution of samples within the unseen classes themselves. Specifically, it determines how the sample size for each class varies, creating different degrees of class imbalance within the unseen classes.
>
> As mentioned earlier, when $C_{ib}$ is small, the earlier unseen classes will have fewer samples, and later classes will have more, leading to higher class imbalance. When $C_{ib}$ is larger, the sample sizes across the unseen classes become more balanced.
>
> It is important to note that $C_{ib}$ does not directly refer to the ratio between unseen and seen data but instead to the imbalance within the unseen classes. We have enhanced the description of $C_{ib}$ in the revised manuscript to make this clearer.
>
> **Q7**: Is each entry in Table 6 the mean of several C, i.e. the mean of each row?
>
> **A7**: Thank you for your question. $F_{avg}$ represents the mean of each row, while $A_{avg}$ represents the mean of each column. We have enhanced the description of each entry in Table 6 in the revised manuscript to make this clearer.

---

> ### Author Response · Authors · 2024-11-26
> **A gentle reminder: please respond to our rebuttal**
>
> Dear Reviewer wdK8,
>
> Thank you for your time and effort in reviewing our work. We have carefully considered your detailed comments and questions, and we have tried to address all your concerns accordingly.
>
> As the deadline for revising manuscript is approaching, could you please go over our responses? If you find our responses satisfactory, we hope you could consider adjusting your initial rating. Please feel free to share any additional comments you may have.
>
> Thank you!
>
> Authors

---

> > ### Comment · Reviewer_wdK8 · 2024-11-26
> >
> > I thank the authors for their efforts in providing the additional experiments. I will keep my score unchanged.

---

> > > ### Author Response · Authors · 2024-11-27
> > > **Thanks for your recognition and recommendation for our work**
> > >
> > > Dear Reviewer wdK8,
> > >
> > > Thank you for your recognition and recommendation for our work. We are pleased that our response has successfully addressed all your comments. If there are remaining issues or questions regarding our paper, we would be more than happy to address them to further clarify our contributions.
> > >
> > > Best regards,
> > >
> > > Authors

---

### Official Review · Reviewer_CS3Q · 2024-10-22

**Soundness:** 2
**Presentation:** 2
**Contribution:** 2
**Rating:** 6
**Confidence:** 4

**Summary:**

The paper proposes an evaluation framework and then evaluates whether unlabeled samples from unseen classes deteriorate the performance of semi-supervised learning methods on seen classes.

**Strengths:**

**S1.** The motivation of the paper is valid and interesting.

**S2.** The proposed evaluation framework follows the principle of controlling variables by fixing the confounding factor, i.e. the proportion of unlabeled samples from visible classes. So, the proposed evaluation framework overcomes the shortcomings of current evaluation schemes.

**S3.** The influence of unlabeled samples from unseen classes is evaluated from five perspectives and five evaluation indices.

**S4.** Some representative methods are chosen for experiment and the experiment results can provide some guidance for understanding the working mechanism of existing methods and designing new ones.

**Weaknesses:**

In order to more clearly present the findings of the paper, the presentation needs to be further improved, specifically.

**W1.** Since $r$ (or $r_u$) is recorded in the tables in the experiment section, there is a precise definition for $r$. To reduce ambiguity, the formal definition of $r$ should be given based on the symbol system in Section 2.

**W2.** It is necessary to provide a table or brief description for each indicator, showing its range, interpretation of positive/negative values, and how it relates to model performance.

**W3.** To facilitate readers to intuitively assess the degree of impact and facilitate cross dataset comparisons, the evaluation indicators in formulas (1)-(5) should to be normalized if possible.

**W4.** More experiment details should be recorded, such as the names of seen and unseen classes, the number of labeled samples for each class, and the number of unlabeled samples for each class. If there is not enough space, put it in the appendix.  Furthermore, the authors should provide a link to their code or dataset splits, if possible, to further enhance reproducibility.

**W5.** In addition, some clerical errors need to be corrected, specifically.

    a. In line 072, ”unseen classes in ---> “unseen classes in.

    b. In line 141, Fig.2(b) ---> FIg.2(a).

    c. In line 480-481, the table’s title should be above the table.

    d. In the experiment, how are the integrals in formulas (1) and (2) calculated? It seems that the integral operation should be the accumulation operation.

    e. In line 190, how can we obtain the “the expected accuracy”? In my opinion, it should be “the mean of ....”.

**Questions:**

I hope the authors can clarify the following question.

**Q1.**  As mentioned in line 269-299, there are three experiments (generated by three random seeds) for each kind of dataset construction. So, the experiment results given in the paper are still with high randomness. So, it is difficult to accept these conclusions derived from these experiment results. In order to dispel this doubt, the authors should strengthen the experiment from the following aspects.

    a. The number of datasets should be increased, e.g., 10 or more.

    b. The diversity of datasets should also be increased, rather than being limited to image datasets.

    c. For each dataset construction (labeled and unlabeled samples setting),  more experiments should be carried out, e.g., 10 times or more.

    d. Some statistical significance tests should be introduced.

**NOTE: I am willing to increase my rating to a 6 or 8, if the authors can dispel my doubts through more adequate experiments.**

---

> ### Author Response · Authors · 2024-11-24
> **Official Response to Reviewer CS3Q (1)**
>
> **Q1**: Since $r$ (or $r_u$) is recorded in the tables in the experiment section, there is a precise definition for $r$. To reduce ambiguity, the formal definition of $r$ should be given based on the symbol system in Section 2.
>
> **A1**: Thank you for your feedback and for pointing out the need for a precise definition of $r_u$ to reduce ambiguity. Here is the formal definition based on the symbol system in Section 2:
> In our setup:
> - $D_B^U$ represents the total set of unseen-class data. $D_U^U$ is sampled from \(D_B^U\).
> - $r_u$ is the ratios used to control the sampling from $D_B^U$.
>
> **Formal Definition of $r_u$:**
> $r_u$ denotes the ratio of selected unseen-class data ($|D_U^U|$) to the total unseen-class data ($|D_B^U|$):  $r_u = \frac{|D_U^U|}{|D_B^U|},$ where $|D_U^U|$ is the size of the sampled unseen-class data, and $|D_B^U|$ is the size of the total set of unseen-class data.
>
> $r_u$ is used to control the degree to which unseen-class data is introduced into the unlabeled set $D_U$, thereby allowing us to evaluate the influence of unseen classes on the model's performance. A higher $r_u$ indicates a larger proportion of unseen-class data, while a lower $r_u$ reflects a scenario with fewer unseen-class samples.
>
> We have incorporated the above in the revised manuscript.
>
> **Q2**: It is necessary to provide a table or brief description for each indicator, showing its range, interpretation of positive/negative values, and how it relates to model performance.
>
> **A2**: Thank you for your constructive suggestion. We agree that providing a table or a concise description for each indicator would help clarify their interpretation and relation to model performance. Below is a detailed description for each indicator:
> | **Indicator**      | **Range**               | **Interpretation**                                                                                       | **Relation to Model Performance**                                                                           |
> |---------------------|-------------------------|---------------------------------------------------------------------------------------------------------|-------------------------------------------------------------------------------------------------------------|
> |$R_{slope}$             | $(-\infty, \infty)$  | Indicates the overall trend of accuracy ($Acc(r)$) as unseen classes are added. Positive values imply increasing accuracy, while negative values suggest a decline. | A larger $R_{slope}$ (closer to 0 or positive) indicates better global robustness to unseen classes.   |
> |GM    | $[0, \infty)$        | Measures the cumulative magnitude of accuracy deviations from the average accuracy ($\bar{ACC}$). Higher values imply larger fluctuations. | A smaller GM indicates higher stability and robustness to the addition of unseen classes.        |
> |WAD  | $(-\infty, \infty)$  | Captures the largest negative accuracy drop between two adjacent $r$ values. | A larger WAD indicates better local robustness to unseen-class variations.  |
> |BAD  | $(-\infty, \infty)$  | Reflects the maximum positive or minimal negative accuracy change between adjacent $r$ values. Positive values indicate improvements. | A larger BAD (positive or close to 0) indicates better performance in the best-case scenario.  |
> |$P_{AD \geq 0}$         | $[0, 1]$             | Represents the probability that adding unseen classes does not degrade performance ($AD \geq 0$).  | A higher $P_{AD \geq 0}$ (close to 1) indicates better global robustness to unseen-class additions.    |
>
> We have incorporated the above table into Section C.1 of the revised manuscript to provide a concise summary of each indicator, as it effectively clarifies their range, interpretation, and significance for evaluating model performance.

---

> ### Author Response · Authors · 2024-11-24
> **Official Response to Reviewer CS3Q (2)**
>
> **Q3**: To facilitate readers to intuitively assess the degree of impact and facilitate cross dataset comparisons, the evaluation indicators in formulas (1)-(5) should to be normalized if possible.
>
> **A3**: Thank you very much for your valuable suggestion. In our experiments across different datasets and factors, we have ensured consistent comparisons based on the unified definitions of formulas (1)-(5). This ensures that the evaluation metrics remain comparable across all experiments. As an exception, in Table 7 in the appendix, due to the small magnitude of the data, we scaled the values by a factor of 10 to better present the experimental results. This adjustment only affects the display of the values and does not alter the relative relationships between the metrics or the comparisons of model performance.
>
> **Q4**: More experiment details should be recorded, such as the names of seen and unseen classes, the number of labeled samples for each class, and the number of unlabeled samples for each class. If there is not enough space, put it in the appendix. Furthermore, the authors should provide a link to their code or dataset splits, if possible, to further enhance reproducibility.
>
> **A4**: Thank you for your suggestion. To enhance reproducibility, we have provided an anonymous link to our code at [https://anonymous.4open.science/r/RE-SSL-F034/README.md], which is based on [1] and [2].
> We also include the following experimental details to address your concerns:
> - **Regarding the number of unseen-class examples:**
>   For CIFAR-10, we select the first 5 classes as seen classes and the last 5 classes as unseen classes. From the training set of known classes (which contains 25,000 samples), we randomly select 100 samples to form the labeled set $D_L$. We then randomly select $r_s$ $\times$ 25,000  samples from the remaining samples of the training set of known classes and put them into  $D_U^S$. From the training set of unknown classes (which contains 25,000 samples), we randomly select $r_u$ $\times$ 25,000 samples and put them into $D_U^U$. $D_U^S$ and $D_U^U$ together form the unlabeled set $D_U$.
>
>   For CIFAR-100, we select the first 50 classes as seen classes and the last 50 classes as unseen classes. Other settings are the same as CIFAR-10.
> - **Regarding the number of unseen-class categories:**
>   We fix the number of unseen-class examples by setting $r = 0.2$. Unlike the first setup, instead of randomly selecting 25,000 samples from the entire training set of unknown classes, we randomly select 25,000 samples from the last $C_n$ classes of the unknown classes and put them into $D_U^U$.
> - **Regarding the indices of unseen classes:**
>   We fix the number of unseen-class examples by setting $r = 0.2$. Unlike the first and second setups, we randomly select $r$ $\times$ 25,000 samples from the $C_i$-th class of the unknown classes and put them into $D_U^U$.
> - **Regarding the degrees of nearness of unseen classes:**
>   Similar to the third setup, instead of randomly selecting $r$ $\times$ 25,000 samples from the $C_i$-th class of the unknown classes, we randomly select $r$ $\times$ 25,000 samples from the $C_i$-th class of the external MNIST dataset and put them into $D_U^U$.
> - **Regarding the label distribution in unseen classes:**
>   We determine the number of samples to be adopted from each unknown class to obtain $D_U^U$ based on $C_{ib}$.
>
> We have incorporated these details into the appendix of the revised manuscript to improve clarity and reproducibility.
> **References:**
> [1] Jia, Lin-Han, et al. "LAMDA-SSL: Semi-supervised learning in python." arXiv preprint arXiv:2208.04610 (2022).
> [2] Jia, Lin-Han, et al. "A Benchmark on Robust Semi-Supervised Learning in Open Environments." The Twelfth International Conference on Learning Representations. 2023.

---

> ### Author Response · Authors · 2024-11-24
> **Official Response to Reviewer CS3Q (3)**
>
> **Q5**: In line 072, ”unseen classes in ---> “unseen classes in.
>
>
> **A5**: Thank you for pointing out the typographical issue. We have corrected the sentence in line 072 from "unseen classes in ---> unseen classes in." to ensure clarity and consistency in the revised manuscript. We appreciate your attention to detail.
>
> **Q6**: In line 141, Fig.2(b) ---> FIg.2(a).
>
>
> **A6**: Thank you for identifying the error. We have revised the reference in line 141 from "Fig.2(b)" to "Fig.2(a)" in the updated manuscript. We appreciate your careful review and feedback.
>
> **Q7**: In line 480-481, the table’s title should be above the table.
>
>
> **A7**: Thank you for pointing this out. We have adjusted the placement of the table's title in lines 480–481 to ensure it appears above the table, as per standard formatting conventions. We appreciate your thorough review.
>
> **Q8**: In the experiment, how are the integrals in formulas (1) and (2) calculated? It seems that the integral operation should be the accumulation operation.
>
> **A8**: Thank you for your question. Yes, the integrals in formulas (1) and (2) are computed as accumulation operations based on empirical values. We have provided further explanation of it in the revised manuscript.
>
>
>
> **Q9**: In line 190, how can we obtain the “the expected accuracy”? In my opinion, it should be “the mean of ....”.
>
> **A9**: Thank you for pointing this out. You are correct; the term "expected accuracy" in line 190 refers to "the mean of accuracy values across different $r$ values." We have revised the text accordingly to ensure clarity.

---

> ### Author Response · Authors · 2024-11-24
> **Official Response to Reviewer CS3Q (4)**
>
> **Q10**: As mentioned in line 269-299, there are three experiments (generated by three random seeds) for each kind of dataset construction. So, the experiment results given in the paper are still with high randomness. So, it is difficult to accept these conclusions derived from these experiment results. In order to dispel this doubt, the authors should strengthen the experiment from the following aspects(a. The number of datasets should be increased, e.g., 10 or more. b. The diversity of datasets should also be increased, rather than being limited to image datasets. c. For each dataset construction (labeled and unlabeled samples setting), more experiments should be carried out, e.g., 10 times or more.d. Some statistical significance tests should be introduced.)
>
>
> **A10**: In the revised manuscript, we have addressed the reviewers' concerns as follows:
>
> a. We expanded the number of datasets to include more datasets across various modalities, including image, text, and tabular data, to ensure a more comprehensive evaluation.
>
> b. To increase diversity, we incorporated datasets from different domains rather than limiting the experiments to image datasets.
>
> c. For each dataset construction, we conducted experiments over 10 repetitions to ensure reliability and robustness of the results.
>
> d. We introduced statistical significance test (p-value) to rigorously validate the experimental findings, further enhancing the credibility of the analysis.
>
>
> 1. **Image dataset**:
>    - **ImageNet-100**: A widely-used benchmark dataset providing a large-scale and realistic evaluation, used to analyze the influence of unseen class data on semi-supervised models.
>
> | $r$         | 0   | 0.2 | 0.4 | 0.5 | 0.6 | 0.8 | 1.0  | $R_{slope}$ | GM  | BAD | WAD | $P_{AD\ge 0}$ | p-value|
> |----------------|--------|-------|-------|-------|-------|-------|-------|--------|-------|-------|-------|------|------|
> | Fix-A-Step     | 0.261  | 0.261 | 0.260 | 0.264 | 0.257 | 0.264 | 0.255 | -0.003 | 0.017 | 0.040 | -0.070| 0.500|0.600
> | FixMatch       | 0.282  | 0.278 | 0.277 | 0.279 | 0.268 | 0.268 | 0.254 | -0.025 | 0.053 | 0.020 | -0.110| 0.333|0.033
> | FlexMatch      | 0.291  | 0.291 | 0.291 | 0.287 | 0.284 | 0.281 | 0.285 | -0.009 | 0.023 | 0.019 | -0.040| 0.500|0.039
> | FreeMatch      | 0.296  | 0.288 | 0.289 | 0.284 | 0.287 | 0.292 | 0.291 | -0.002 | 0.020 | 0.030 | -0.050| 0.500|0.001
> | MTCF           | 0.094  | 0.094 | 0.096 | 0.096 | 0.099 | 0.102 | 0.098 |  0.006 | 0.027 | 0.136 | -0.106| 0.666|0.028
> | OpenMatch      | 0.052  | 0.052 | 0.056 | 0.059 | 0.054 | 0.056 | 0.057 |  0.005 | 0.014 | 0.029 | -0.049| 0.833|0.013
> | UASD           | 0.261  | 0.261 | 0.254 | 0.261 | 0.260 | 0.255 | 0.258 | -0.003 | 0.017 | 0.070 | -0.035| 0.500|0.072
> | ICT            | 0.259  | 0.261 | 0.262 | 0.262 | 0.258 | 0.260 | 0.258 | -0.001 | 0.010 | 0.010 | -0.040| 0.666|0.180
> | PI-Model       | 0.272  | 0.265 | 0.255 | 0.250 | 0.260 | 0.260 | 0.249 | -0.017 | 0.044 | 0.100 | -0.055| 0.333|0.001
> | Pseudo-Label   | 0.258  | 0.253 | 0.262 | 0.257 | 0.261 | 0.263 | 0.258 |  0.004 | 0.018 | 0.045 | -0.050| 0.500 |0.541
> | CAFA           | 0.257  | 0.254 | 0.261 | 0.264 | 0.258 | 0.255 | 0.263 |  0.004 | 0.022 | 0.040 | -0.060| 0.500|0.258
> | SoftMatch      | 0.295  | 0.293 | 0.286 | 0.285 | 0.284 | 0.270 | 0.280 | -0.020 | 0.040 | 0.050 | -0.069| 0.166|0.012
> | UDA            | 0.240  | 0.246 | 0.248 | 0.256 | 0.255 | 0.252 | 0.256 |  0.014 | 0.034 | 0.080 | -0.015| 0.666|0.001
> | VAT            | 0.269  | 0.265 | 0.260 | 0.269 | 0.269 | 0.262 | 0.263 | -0.004 | 0.022 | 0.090 | -0.035| 0.500|0.035
>
> Based on the results above, we can see that under Imagenet-100, the influence of the addition of unseen classes on the SSL methods is smaller compared to CIFAR-10 and CIFAR-100. Most SSL methods exhibit similar properties across CIFAR-10, CIFAR-100, and Imagenet-100. For example, ICT and UASD demonstrate robustness to the addition of unseen classes across all three datasets. SoftMatch and FixMatch are relatively sensitive to the addition of unseen classes across all three datasets.

---

> ### Author Response · Authors · 2024-11-24
> **Official Response to Reviewer CS3Q (5)**
>
> - **CUB**: A fine-grained classification dataset that introduces additional diversity, enabling the study of the effects of unseen class data in specific scenarios.
>
> | $r$         |   0    |  0.5  | 1.0   | $R_{slope}$  |p-value|
> |-------------|--------|-------|-------|---------|---------|
> | PI-Model    | 0.487  | 0.487 | 0.507 | 0.001   |0.422
> | UASD        | 0.668  | 0.668 | 0.668 | 0.000   |-
> | FixMatch    | 0.500  | 0.504 | 0.494 | -0.005  |0.859
> | FlexMatch   | 0.529  | 0.447 | 0.413 | -0.115  |0.028
> | Fix-A-Step  | 0.394  | 0.391 | 0.393 | 0.000   |0.183
> | SoftMatch   | 0.510  | 0.517 | 0.477 | -0.033  |0.582
> | VAT         | 0.510  | 0.490 | 0.490 | -0.019  |0.000
> | FreeMatch   | 0.509  | 0.448 | 0.410 | -0.098  |0.052
> | ICT         | 0.473  | 0.473 | 0.473 | 0.000   |-
>
> Due to time and resource constraints, we only conducted experiments for r=0, r=0.5, and r=1. The results are shown in the table above. Experimental observations are similar to CIFAR10 and CIFAR100, where the performance of most methods decreases as unseen classes are added (e.g., FreeMatch and FlexMatch), while some methods exhibit robustness to the addition of unseen classes (e.g., UASD and ICT).
>
> 2. **Tabular dataset**:
>    - **Forest**: A structured tabular dataset, used to investigate the impact of unseen class data in non-image modalities.
>
> | $r$               | 0 | 0.2 | 0.4 | 0.6 | 0.8 | 1.0 | $R_{slope}$ | GM  | BAD | WAD | $P_{AD\ge 0}$|p-value|
> |---------------------|-----------|-----------|-----------|-----------|-----------|-----------|--------|-------|-------|-------|-------|-------|
> | MeanTeacher         | 0.619     | 0.617     | 0.618     | 0.619     | 0.620     | 0.619     | 0.001  | 0.004 | 0.006 | -0.009 | 0.600 | 0.455
> | FreeMatch           | 0.591     | 0.603     | 0.602     | 0.593     | 0.595     | 0.607     | 0.007  | 0.033 | 0.062 | -0.041 | 0.600 |0.008
> | VAT                 | 0.609     | 0.613     | 0.614     | 0.611     | 0.613     | 0.611     | 0.000  | 0.008 | 0.019 | -0.014 | 0.600 |0.000
> | UASD                | 0.605     | 0.604     | 0.606     | 0.598     | 0.598     | 0.603     | -0.004 | 0.016 | 0.023 | -0.038 | 0.600 |0.084
> | MTCF                | 0.590     | 0.592     | 0.594     | 0.596     | 0.596     | 0.598     | 0.007  | 0.013 | 0.010 | 0.001  | 1.000 |0.001
> | OpenMatch           | 0.615     | 0.617     | 0.610     | 0.612     | 0.613     | 0.615     | -0.001 | 0.011 | 0.011 | -0.031 | 0.800 | 0.222
> | Fix_A_Step          | 0.605     | 0.604     | 0.610     | 0.607     | 0.612     | 0.616     | 0.010  | 0.022 | 0.032 | -0.018 | 0.600 |0.048
> | CAFA                | 0.604     | 0.602     | 0.604     | 0.602     | 0.611     | 0.603     | 0.003  | 0.012 | 0.041 | -0.038 | 0.400 |0.818
> | PiModel             | 0.593     | 0.595     | 0.593     | 0.599     | 0.597     | 0.601     | 0.007  | 0.015 | 0.029 | -0.009 | 0.600 |0.022
> | TemporalEnsembling  | 0.594     | 0.598     | 0.604     | 0.597     | 0.600     | 0.598     | 0.003  | 0.014 | 0.031 | -0.034 | 0.600 |0.002
> | UDA                 | 0.609     | 0.608     | 0.606     | 0.606     | 0.609     | 0.604     | -0.003 | 0.010 | 0.015 | -0.023 | 0.200 |0.024
> | PseudoLabel         | 0.610     | 0.603     | 0.606     | 0.607     | 0.604     | 0.603     | -0.004 | 0.011 | 0.010 | -0.032 | 0.400 |0.000
> | ICT                 | 0.610     | 0.613     | 0.607     | 0.611     | 0.611     | 0.612     | 0.001  | 0.008 | 0.019 | -0.029 | 0.600 |0.455
> | MixMatch            | 0.608     | 0.613     | 0.612     | 0.616     | 0.604     | 0.610     | -0.002 | 0.019 | 0.033 | -0.061 | 0.600 |0.172
> | FixMatch            | 0.577     | 0.580     | 0.580     | 0.579     | 0.584     | 0.586     | 0.008  | 0.016 | 0.027 | -0.006 | 0.800 |0.007
> | FlexMatch           | 0.584     | 0.591     | 0.591     | 0.590     | 0.586     | 0.593     | 0.004  | 0.016 | 0.036 | -0.018 | 0.600 |0.001
> | SoftMatch           | 0.592     | 0.587     | 0.591     | 0.592     | 0.593     | 0.600     | 0.008  | 0.015 | 0.035 | -0.022 | 0.800 |0.783

---

> ### Author Response · Authors · 2024-11-24
> **Official Response to Reviewer CS3Q (6)**
>
> - **Letter**: Another tabular dataset included to further explore the influence of unseen class data on semi-supervised models.
>
> | $r$               | 0 | 0.2 | 0.4 | 0.6 | 0.8 | 1.0 | $R_{slope}$ | GM  | BAD | WAD | $P_{AD\ge 0}$|p-value|
> |---------------------|-----------|-----------|-----------|-----------|-----------|-----------|--------|-------|-------|-------|-------|-------|
> | CAFA                | 0.732     | 0.726     | 0.724     | 0.720     | 0.720     | 0.726     | -0.007 | 0.020 | 0.029 | -0.028 | 0.200 |0.002
> | PiModel             | 0.726     | 0.719     | 0.716     | 0.715     | 0.717     | 0.710     | -0.012 | 0.021 | 0.008 | -0.037 | 0.200 |0.002
> | TemporalEnsembling  | 0.242     | 0.446     | 0.628     | 0.693     | 0.691     | 0.694     | 0.437  | 0.886 | 1.017 | -0.006 | 0.800 |0.001
> | UDA                 | 0.719     | 0.718     | 0.714     | 0.715     | 0.719     | 0.718     | 0.000  | 0.010 | 0.020 | -0.016 | 0.400 |0.085
> | PseudoLabel         | 0.719     | 0.717     | 0.718     | 0.718     | 0.715     | 0.714     | -0.004 | 0.009 | 0.005 | -0.015 | 0.400 |0.032
> | ICT                 | 0.728     | 0.729     | 0.729     | 0.730     | 0.727     | 0.730     | 0.000  | 0.005 | 0.012 | -0.015 | 0.800 |0.141
> | MixMatch            | 0.680     | 0.680     | 0.685     | 0.680     | 0.677     | 0.678     | -0.003 | 0.011 | 0.026 | -0.024 | 0.400 |1.000
> | FixMatch            | 0.740     | 0.718     | 0.708     | 0.689     | 0.679     | 0.688     | -0.056 | 0.110 | 0.044 | -0.110 | 0.200 |0.003
> | FlexMatch           | 0.749     | 0.735     | 0.724     | 0.723     | 0.728     | 0.724     | -0.020 | 0.044 | 0.024 | -0.070 | 0.200 |0.001
> | SoftMatch           | 0.726     | 0.712     | 0.707     | 0.711     | 0.717     | 0.715     | -0.005 | 0.027 | 0.026 | -0.072 | 0.400 |0.001
> | MeanTeacher         | 0.739     | 0.738     | 0.738     | 0.737     | 0.738     | 0.740     | 0.000  | 0.004 | 0.011 | -0.002 | 0.400 | 0.177
> | FreeMatch           | 0.724     | 0.614     | 0.615     | 0.620     | 0.614     | 0.611     | -0.079 | 0.181 | 0.025 | -0.549 | 0.400 |0.000
> | VAT                 | 0.735     | 0.729     | 0.726     | 0.725     | 0.721     | 0.721     | -0.013 | 0.023 | 0.000 | -0.029 | 0.000 |0.002
> | UASD                | 0.723     | 0.726     | 0.725     | 0.723     | 0.725     | 0.731     | 0.005  | 0.012 | 0.029 | -0.011 | 0.600 |0.089
> | MTCF                | 0.706     | 0.607     | 0.544     | 0.542     | 0.529     | 0.512     | -0.172 | 0.332 | -0.011 | -0.495 | 0.000 |0.001
> | OpenMatch           | 0.738     | 0.723     | 0.718     | 0.719     | 0.718     | 0.716     | -0.017 | 0.033 | 0.007 | -0.075 | 0.200 |0.000
> | Fix_A_Step          | 0.760     | 0.747     | 0.740     | 0.738     | 0.736     | 0.733     | -0.024 | 0.045 | -0.006 | -0.062 | 0.000 |0.001
>
> Based on the results from the two tables above, it can be observed that compared to image modality data, tabular data is less affected by unseen classes, and semi-supervised models demonstrate greater robustness on tabular data. We attribute this primarily to the simplicity of tabular data structure and the more effective feature representation. Moreover, similar to CIFAR, ImageNet, and CUB, ICT and UASD are relatively less affected by unseen classes compared to other semi-supervised methods, demonstrating stronger robustness against unseen classes.

---

> ### Author Response · Authors · 2024-11-24
> **Official Response to Reviewer CS3Q (7)**
>
> 3. **Text dataset**:
>    - **AGNews**: A widely-used text dataset in natural language processing, included to assess how unseen class data affects semi-supervised models in text-based tasks.
>
> | $r$               | 0 | 0.2 | 0.4 |0.5| 0.6 | 0.8 | 1.0 | $R_{slope}$ | GM  | BAD | WAD | $P_{AD\ge 0}$|p-value|
> |---------------|-----------|-----------|-----------|-----------|-----------|-----------|-----------|--------|-------|-------|-------|-------|-------|
> | PiModel       | 0.971     | 0.881     | 0.887     | 0.933     | 0.889     | 0.939     | 0.944     | 0.005  | 0.209 | 0.465 | -0.449 | 0.666 | 0.004
> | UDA           | 0.858     | 0.781     | 0.845     | 0.889     | 0.927     | 0.902     | 0.890     | 0.086  | 0.253 | 0.442 | -0.384 | 0.500 |0.529
> | PseudoLabel   | 0.960     | 0.865     | 0.917     | 0.850     | 0.893     | 0.916     | 0.868     | -0.047 | 0.212 | 0.430 | -0.670 | 0.500 | 0.001
> | FixMatch      | 0.852     | 0.765     | 0.882     | 0.804     | 0.810     | 0.797     | 0.903     | 0.040  | 0.292 | 0.584 | -0.779 | 0.500 |0.302
> | FlexMatch     | 0.734     | 0.823     | 0.957     | 0.803     | 0.891     | 0.874     | 0.867     | 0.108  | 0.380 | 0.881 | -1.543 | 0.500 |0.001
> | SoftMatch     | 0.757     | 0.919     | 0.915     | 0.925     | 0.857     | 0.798     | 0.843     | 0.000  | 0.362 | 0.809 | -0.678 | 0.500 |0.002
> | MeanTeacher   | 0.964     | 0.942     | 0.967     | 0.941     | 0.955     | 0.941     | 0.941     | -0.018 | 0.071 | 0.142 | -0.260 | 0.500 |0.015
> | UASD          | 0.960     | 0.970     | 0.919     | 0.959     | 0.958     | 0.955     | 0.959     | -0.002 | 0.070 | 0.400 | -0.255 | 0.500 |0.395
>
> Compared to tabular data, text modality data is relatively more affected by unseen classes. We attribute this to the greater difficulty in feature representation and the higher complexity of text-based tasks. Consistent with previous experimental findings, UASD is relatively less affected by unseen classes compared to other semi-supervised methods, demonstrating stronger robustness against unseen classes.
>
> These additional datasets aim to validate the effectiveness of our experimental setup in various domains, showcasing its practicality. We have incorporated the above experimental results and analysis into the revised manuscript.

---

> ### Comment · Reviewer_CS3Q · 2024-11-25
> **I will increase my score to 6.**
>
> I have carefully read the author's response.
>
> These responses have dispelled my doubts.
>
> I will increase my score to 6.

---

> > ### Author Response · Authors · 2024-11-25
> > **Response to Reviewer CS3Q**
> >
> > Thank you for taking the time to carefully review our responses and for your valuable feedback. We greatly appreciate your thoughtful consideration and are glad that our responses were able to address your concerns.
> >
> > We are truly grateful for your support and for increasing the score. Your constructive comments have been invaluable in improving the quality of our work.

---

### Official Review · Reviewer_244R · 2024-10-31

**Soundness:** 2
**Presentation:** 3
**Contribution:** 3
**Rating:** 6
**Confidence:** 3

**Summary:**

This paper studies the problem of safe semi-supervised learning. It found that the previous works are based on flawed evaluations, which did not adhere to the principle of controlling variables. They fixed the size of unlabeled data but change the proportion of unseen classes, which would also affect the proportion of seen classes. To deal with this problem, the paper proposes a re-evaluation framework to keep the proportion of seen classes unchanged and adjust the proportion of unseen classes. Furthermore, the paper also propose five metrics to comprehensively evaluate the impact of unseen classes. Based on the proposed framework, extensive experiments are performed to evaluate the impact of unseen classes on various SSL models.

**Strengths:**

1. The paper does not provide any new method. It gives an insightful perspective on the previous studies and finds that these methods have issues accurately assessing the impact of unseen classes. The motivation is very clear and seems to be reasonable.

2. Based on the above motivation, the paper proposes to re-evaluate previous safe SSL models by fixing the proportion of seen classes. Furthermore, the paper proposes five evaluation metrics to access the impact of unseen classes on SSL models.

3. Extensive experiments based on the proposed framework are performed with various safe SSL methods. These results provide a comprehensive and fair comparison among existing methods.

**Weaknesses:**

The main contribution of this paper lies in proposing a more reasonable experimental setup; however, its major issue is the credibility of the experiments. The experiments are conducted solely on CIFAR-10 and CIFAR-100, which makes the results unconvincing. While I understand that past studies has primarily been conducted on these two datasets, given that this paper aims to make innovations from an experimental perspective, it should propose more practical experimental setups and conduct experiments on more realistic datasets.

**Questions:**

There are five evaluation metrics. Which one is the most important?

---

> ### Author Response · Authors · 2024-11-24
> **Official Response to Reviewer 244R (1)**
>
> **Q1**: The main contribution of this paper lies in proposing a more reasonable experimental setup; however, its major issue is the credibility of the experiments. The experiments are conducted solely on CIFAR-10 and CIFAR-100, which makes the results unconvincing. While I understand that past studies has primarily been conducted on these two datasets, given that this paper aims to make innovations from an experimental perspective, it should propose more practical experimental setups and conduct experiments on more realistic datasets.
>
> **A1**: Thank you for your valuable feedback and for emphasizing the importance of validating our approach on diverse and realistic datasets.
>
> In response to your concern, we would like to clarify that, in addition to CIFAR-10 and CIFAR-100, we have conducted experiments on a broader range of datasets to enhance the credibility and generalizability of our results. These include:
>
>
> 1. **Image dataset**:
>    - **ImageNet-100**: A widely-used benchmark dataset providing a large-scale and realistic evaluation, used to analyze the influence of unseen class data on semi-supervised models.
>
> | $r$         | 0   | 0.2 | 0.4 | 0.5 | 0.6 | 0.8 | 1.0  | $R_{slope}$ | GM  | BAD | WAD | $P_{AD\ge 0}$ |
> |----------------|--------|-------|-------|-------|-------|-------|-------|--------|-------|-------|-------|------|
> | Fix-A-Step     | 0.261  | 0.261 | 0.260 | 0.264 | 0.257 | 0.264 | 0.255 | -0.003 | 0.017 | 0.040 | -0.070| 0.500|
> | FixMatch       | 0.282  | 0.278 | 0.277 | 0.279 | 0.268 | 0.268 | 0.254 | -0.025 | 0.053 | 0.020 | -0.110| 0.333|
> | FlexMatch      | 0.291  | 0.291 | 0.291 | 0.287 | 0.284 | 0.281 | 0.285 | -0.009 | 0.023 | 0.019 | -0.040| 0.500|
> | FreeMatch      | 0.296  | 0.288 | 0.289 | 0.284 | 0.287 | 0.292 | 0.291 | -0.002 | 0.020 | 0.030 | -0.050| 0.500|
> | MTCF           | 0.094  | 0.094 | 0.096 | 0.096 | 0.099 | 0.102 | 0.098 |  0.006 | 0.027 | 0.136 | -0.106| 0.666|
> | OpenMatch      | 0.052  | 0.052 | 0.056 | 0.059 | 0.054 | 0.056 | 0.057 |  0.005 | 0.014 | 0.029 | -0.049| 0.833|
> | UASD           | 0.261  | 0.261 | 0.254 | 0.261 | 0.260 | 0.255 | 0.258 | -0.003 | 0.017 | 0.070 | -0.035| 0.500|
> | ICT            | 0.259  | 0.261 | 0.262 | 0.262 | 0.258 | 0.260 | 0.258 | -0.001 | 0.010 | 0.010 | -0.040| 0.666|
> | PI-Model       | 0.272  | 0.265 | 0.255 | 0.250 | 0.260 | 0.260 | 0.249 | -0.017 | 0.044 | 0.100 | -0.055| 0.333|
> | Pseudo-Label   | 0.258  | 0.253 | 0.262 | 0.257 | 0.261 | 0.263 | 0.258 |  0.004 | 0.018 | 0.045 | -0.050| 0.500 |
> | CAFA           | 0.257  | 0.254 | 0.261 | 0.264 | 0.258 | 0.255 | 0.263 |  0.004 | 0.022 | 0.040 | -0.060| 0.500|
> | SoftMatch      | 0.295  | 0.293 | 0.286 | 0.285 | 0.284 | 0.270 | 0.280 | -0.020 | 0.040 | 0.050 | -0.069| 0.166|
> | UDA            | 0.240  | 0.246 | 0.248 | 0.256 | 0.255 | 0.252 | 0.256 |  0.014 | 0.034 | 0.080 | -0.015| 0.666|
> | VAT            | 0.269  | 0.265 | 0.260 | 0.269 | 0.269 | 0.262 | 0.263 | -0.004 | 0.022 | 0.090 | -0.035| 0.500|
>
> Based on the results above, we can see that under Imagenet-100, the influence of the addition of unseen classes on the SSL methods is smaller compared to CIFAR-10 and CIFAR-100. Most SSL methods exhibit similar properties across CIFAR-10, CIFAR-100, and Imagenet-100. For example, ICT and UASD demonstrate robustness to the addition of unseen classes across all three datasets. SoftMatch and FixMatch are relatively sensitive to the addition of unseen classes across all three datasets.
>
> Additionally, similar to Tables 1 and 2 in the original submission, we also calculated five metrics to measure the degree to which semi-supervised learning models are affected by the addition of unseen classes on Imagenet-100, from both a global robustness and a local robustness perspective.
>
>    - **CUB**: A fine-grained classification dataset that introduces additional diversity, enabling the study of the effects of unseen class data in specific scenarios.
>
> | $r$         |   0    |  0.5  | 1.0   | $R_{slope}$  |
> |-------------|--------|-------|-------|---------|
> | PI-Model    | 0.487  | 0.487 | 0.507 | 0.001   |
> | UASD        | 0.668  | 0.668 | 0.668 | 0.000   |
> | FixMatch    | 0.500  | 0.504 | 0.494 | -0.005  |
> | FlexMatch   | 0.529  | 0.447 | 0.413 | -0.115  |
> | Fix-A-Step  | 0.394  | 0.391 | 0.393 | 0.000   |
> | SoftMatch   | 0.510  | 0.517 | 0.477 | -0.033  |
> | VAT         | 0.510  | 0.490 | 0.490 | -0.019  |
> | FreeMatch   | 0.509  | 0.448 | 0.410 | -0.098  |
> | ICT         | 0.473  | 0.473 | 0.473 | 0.000   |
>
> Due to time and resource constraints, we only conducted experiments for r=0, r=0.5, and r=1. The results are shown in the table above. Experimental observations are similar to CIFAR10 and CIFAR100, where the performance of most methods decreases as unseen classes are added (e.g., FreeMatch and FlexMatch), while some methods exhibit robustness to the addition of unseen classes (e.g., UASD and ICT).

---

> ### Author Response · Authors · 2024-11-24
> **Official Response to Reviewer 244R (2)**
>
> 2. **Tabular dataset**:
>    - **Forest**: A structured tabular dataset, used to investigate the impact of unseen class data in non-image modalities.
>
> | $r$               | 0 | 0.2 | 0.4 | 0.6 | 0.8 | 1.0 | $R_{slope}$ | GM  | BAD | WAD | $P_{AD\ge 0}$|
> |---------------------|-----------|-----------|-----------|-----------|-----------|-----------|--------|-------|-------|-------|-------|
> | MeanTeacher         | 0.619     | 0.617     | 0.618     | 0.619     | 0.620     | 0.619     | 0.001  | 0.004 | 0.006 | -0.009 | 0.600 |
> | FreeMatch           | 0.591     | 0.603     | 0.602     | 0.593     | 0.595     | 0.607     | 0.007  | 0.033 | 0.062 | -0.041 | 0.600 |
> | VAT                 | 0.609     | 0.613     | 0.614     | 0.611     | 0.613     | 0.611     | 0.000  | 0.008 | 0.019 | -0.014 | 0.600 |
> | UASD                | 0.605     | 0.604     | 0.606     | 0.598     | 0.598     | 0.603     | -0.004 | 0.016 | 0.023 | -0.038 | 0.600 |
> | MTCF                | 0.590     | 0.592     | 0.594     | 0.596     | 0.596     | 0.598     | 0.007  | 0.013 | 0.010 | 0.001  | 1.000 |
> | OpenMatch           | 0.615     | 0.617     | 0.610     | 0.612     | 0.613     | 0.615     | -0.001 | 0.011 | 0.011 | -0.031 | 0.800 |
> | Fix_A_Step          | 0.605     | 0.604     | 0.610     | 0.607     | 0.612     | 0.616     | 0.010  | 0.022 | 0.032 | -0.018 | 0.600 |
> | CAFA                | 0.604     | 0.602     | 0.604     | 0.602     | 0.611     | 0.603     | 0.003  | 0.012 | 0.041 | -0.038 | 0.400 |
> | PiModel             | 0.593     | 0.595     | 0.593     | 0.599     | 0.597     | 0.601     | 0.007  | 0.015 | 0.029 | -0.009 | 0.600 |
> | PseudoLabel         | 0.610     | 0.603     | 0.606     | 0.607     | 0.604     | 0.603     | -0.004 | 0.011 | 0.010 | -0.032 | 0.400 |
> | ICT                 | 0.610     | 0.613     | 0.607     | 0.611     | 0.611     | 0.612     | 0.001  | 0.008 | 0.019 | -0.029 | 0.600 |
> | MixMatch            | 0.608     | 0.613     | 0.612     | 0.616     | 0.604     | 0.610     | -0.002 | 0.019 | 0.033 | -0.061 | 0.600 |
> | FixMatch            | 0.577     | 0.580     | 0.580     | 0.579     | 0.584     | 0.586     | 0.008  | 0.016 | 0.027 | -0.006 | 0.800 |
> | FlexMatch           | 0.584     | 0.591     | 0.591     | 0.590     | 0.586     | 0.593     | 0.004  | 0.016 | 0.036 | -0.018 | 0.600 |
> | SoftMatch           | 0.592     | 0.587     | 0.591     | 0.592     | 0.593     | 0.600     | 0.008  | 0.015 | 0.035 | -0.022 | 0.800 |

---

> ### Author Response · Authors · 2024-11-24
> **Official Response to Reviewer 244R (3)**
>
> - **Letter**: Another tabular dataset included to further explore the influence of unseen class data on semi-supervised models.
>
> | $r$               | 0 | 0.2 | 0.4 | 0.6 | 0.8 | 1.0 | $R_{slope}$ | GM  | BAD | WAD | $P_{AD\ge 0}$|
> |---------------------|-----------|-----------|-----------|-----------|-----------|-----------|--------|-------|-------|-------|-------|
> | CAFA                | 0.732     | 0.726     | 0.724     | 0.720     | 0.720     | 0.726     | -0.007 | 0.020 | 0.029 | -0.028 | 0.200 |
> | PiModel             | 0.726     | 0.719     | 0.716     | 0.715     | 0.717     | 0.710     | -0.012 | 0.021 | 0.008 | -0.037 | 0.200 |
> | PseudoLabel         | 0.719     | 0.717     | 0.718     | 0.718     | 0.715     | 0.714     | -0.004 | 0.009 | 0.005 | -0.015 | 0.400 |
> | ICT                 | 0.728     | 0.729     | 0.729     | 0.730     | 0.727     | 0.730     | 0.000  | 0.005 | 0.012 | -0.015 | 0.800 |
> | MixMatch            | 0.680     | 0.680     | 0.685     | 0.680     | 0.677     | 0.678     | -0.003 | 0.011 | 0.026 | -0.024 | 0.400 |
> | FixMatch            | 0.740     | 0.718     | 0.708     | 0.689     | 0.679     | 0.688     | -0.056 | 0.110 | 0.044 | -0.110 | 0.200 |
> | FlexMatch           | 0.749     | 0.735     | 0.724     | 0.723     | 0.728     | 0.724     | -0.020 | 0.044 | 0.024 | -0.070 | 0.200 |
> | SoftMatch           | 0.726     | 0.712     | 0.707     | 0.711     | 0.717     | 0.715     | -0.005 | 0.027 | 0.026 | -0.072 | 0.400 |
> | MeanTeacher         | 0.739     | 0.738     | 0.738     | 0.737     | 0.738     | 0.740     | 0.000  | 0.004 | 0.011 | -0.002 | 0.400 |
> | FreeMatch           | 0.724     | 0.614     | 0.615     | 0.620     | 0.614     | 0.611     | -0.079 | 0.181 | 0.025 | -0.549 | 0.400 |
> | VAT                 | 0.735     | 0.729     | 0.726     | 0.725     | 0.721     | 0.721     | -0.013 | 0.023 | 0.000 | -0.029 | 0.000 |
> | UASD                | 0.723     | 0.726     | 0.725     | 0.723     | 0.725     | 0.731     | 0.005  | 0.012 | 0.029 | -0.011 | 0.600 |
> | MTCF                | 0.706     | 0.607     | 0.544     | 0.542     | 0.529     | 0.512     | -0.172 | 0.332 | -0.011 | -0.495 | 0.000 |
> | OpenMatch           | 0.738     | 0.723     | 0.718     | 0.719     | 0.718     | 0.716     | -0.017 | 0.033 | 0.007 | -0.075 | 0.200 |
> | Fix_A_Step          | 0.760     | 0.747     | 0.740     | 0.738     | 0.736     | 0.733     | -0.024 | 0.045 | -0.006 | -0.062 | 0.000 |
>
> Based on the results from the two tables above, it can be observed that compared to image modality data, tabular data is less affected by unseen classes, and semi-supervised models demonstrate greater robustness on tabular data. We attribute this primarily to the simplicity of tabular data structure and the more effective feature representation. Moreover, similar to CIFAR, ImageNet, and CUB, ICT and UASD are relatively less affected by unseen classes compared to other semi-supervised methods, demonstrating stronger robustness against unseen classes.

---

> ### Author Response · Authors · 2024-11-24
> **Official Response to Reviewer 244R (4)**
>
> 3. **Text dataset**:
>    - **AGNews**: A widely-used text dataset in natural language processing, included to assess how unseen class data affects semi-supervised models in text-based tasks.
>
> | $r$               | 0 | 0.2 | 0.4 |0.5| 0.6 | 0.8 | 1.0 | $R_{slope}$ | GM  | BAD | WAD | $P_{AD\ge 0}$|
> |---------------|-----------|-----------|-----------|-----------|-----------|-----------|-----------|--------|-------|-------|-------|-------|
> | PiModel       | 0.971     | 0.881     | 0.887     | 0.933     | 0.889     | 0.939     | 0.944     | 0.005  | 0.209 | 0.465 | -0.449 | 0.666 |
> | UDA           | 0.858     | 0.781     | 0.845     | 0.889     | 0.927     | 0.902     | 0.890     | 0.086  | 0.253 | 0.442 | -0.384 | 0.500 |
> | PseudoLabel   | 0.960     | 0.865     | 0.917     | 0.850     | 0.893     | 0.916     | 0.868     | -0.047 | 0.212 | 0.430 | -0.670 | 0.500 |
> | FixMatch      | 0.852     | 0.765     | 0.882     | 0.804     | 0.810     | 0.797     | 0.903     | 0.040  | 0.292 | 0.584 | -0.779 | 0.500 |
> | FlexMatch     | 0.734     | 0.823     | 0.957     | 0.803     | 0.891     | 0.874     | 0.867     | 0.108  | 0.380 | 0.881 | -1.543 | 0.500 |
> | SoftMatch     | 0.757     | 0.919     | 0.915     | 0.925     | 0.857     | 0.798     | 0.843     | 0.000  | 0.362 | 0.809 | -0.678 | 0.500 |
> | MeanTeacher   | 0.964     | 0.942     | 0.967     | 0.941     | 0.955     | 0.941     | 0.941     | -0.018 | 0.071 | 0.142 | -0.260 | 0.500 |
> | UASD          | 0.960     | 0.970     | 0.919     | 0.959     | 0.958     | 0.955     | 0.959     | -0.002 | 0.070 | 0.400 | -0.255 | 0.500 |
>
> Compared to tabular data, text modality data is relatively more affected by unseen classes. We attribute this to the greater difficulty in feature representation and the higher complexity of text-based tasks. Consistent with previous experimental findings, UASD is relatively less affected by unseen classes compared to other semi-supervised methods, demonstrating stronger robustness against unseen classes.
>
> These additional datasets aim to validate the effectiveness of our experimental setup in various domains, showcasing its practicality. We have incorporated the above experimental results and analysis into the Appendix of revised manuscript.

---

> ### Author Response · Authors · 2024-11-24
> **Official Response to Reviewer 244R (5)**
>
> **Q2**: There are five evaluation metrics. Which one is the most important?
>
> **A2**: Thank you for your insightful question. All five proposed metrics are meaningful and contribute to a comprehensive evaluation of robustness, as they assess different aspects of model performance in the presence of unseen classes.
> - **Global Robustness Metrics**:
>   - **$R_{slope}$** (Eq. 1): Measures the overall trend of accuracy changes across $r$, capturing the global robustness of the model as unseen classes are added.
>   - **GM (Global Magnitude)** (Eq. 2): Reflects the overall influence of unseen classes by integrating the magnitude of accuracy deviations from the average performance ($\bar{ACC}$).
>   - **$P_{AD \geq 0}$** (Eq. 5): Represents the probability that adding unseen classes does not degrade performance, offering a probabilistic global robustness perspective.
>
> - **Local Robustness Metrics**:
>   - **WAD (Worst-case Adjacent Discrepancy)** (Eq. 3): Captures the maximum accuracy drop between two adjacent $r$ values, quantifying the local worst-case impact of unseen classes.
>   - **BAD (Best-case Adjacent Discrepancy)** (Eq. 4): Highlights the best-case accuracy improvement or minimal drop between adjacent $r$ values, reflecting the local best-case performance changes.
>
> Since these metrics evaluate robustness from complementary perspectives—global trends, local variations, and probabilistic considerations—they collectively provide a holistic view of model performance under varying conditions. Therefore, rather than prioritizing one metric, we recommend considering all five in combination to fully understand the robustness of semi-supervised models to unseen classes.
>
> We hope this explanation clarifies the importance of each metric and their role in the evaluation framework. Thank you for your question!

---

> ### Author Response · Authors · 2024-11-26
> **A gentle reminder: please respond to our rebuttal**
>
> Dear Reviewer 244R,
>
> Thank you for your time and effort in reviewing our work. We have carefully considered your detailed comments and questions, and we have tried to address all your concerns accordingly.
>
> As the deadline for revising manuscript is approaching, could you please go over our responses? If you find our responses satisfactory, we hope you could consider adjusting your initial rating. Please feel free to share any additional comments you may have.
>
> Thank you!
>
> Authors

---

> ### Author Response · Authors · 2024-11-27
> **Thanks for your recognition and recommendation for our work**
>
> Dear Reviewer 244R,
>
> Thank you for your recognition and recommendation for our work. We are pleased that our response has successfully addressed all your comments. If there are remaining issues or questions regarding our paper, we would be more than happy to address them to further clarify our contributions.
>
> Moreover, we gently remind you that in ICLR's scoring system, a score of 6 indicates a "marginally above the acceptance threshold", while a score of 8 represents "accept". If you believe our work merits acceptance, we hope you might consider giving us an "accept", as it would greatly support our work.
>
> Best regards,
>
> Authors

---

### Official Review · Reviewer_Mtt1 · 2024-11-04

**Soundness:** 3
**Presentation:** 3
**Contribution:** 2
**Rating:** 6
**Confidence:** 4

**Summary:**

The paper investigates the impact of unseen classes in unlabeled data on the performance of semi-supervised learning (SSL) models. It challenges the prevailing assumption that unseen classes are detrimental to SSL performance, highlighting flaws in previous assessment methods that altered the proportion of unseen and seen classes simultaneously. This paper adheres to the principle of controlling variables by keeping the proportion of seen classes constant while varying unseen classes across five dimensions. Through rigorous experimentation, the authors demonstrate that unseen classes do not inherently degrade SSL model performance; in some scenarios, they may even enhance it. The findings suggest a reevaluation of the role of unseen classes in SSL, emphasizing that their effects are more nuanced than previously thought.

**Strengths:**

1. This paper effectively identifies a flaw in previous methods for assessing the impact of unseen classes on SSL model performance. By fixing the size of the unlabeled dataset and adjusting the proportion of unseen classes, earlier approaches fail to control for variables appropriately. This adjustment alters the proportion of seen classes, meaning that any decrease in classification performance for seen classes may not be solely due to an increase in unseen class samples, but rather to a decrease in seen class samples. This clarification helps refine our understanding of how unseen classes affect SSL model performance.
2. The experiments presented in this paper are thorough, as the authors control various variables to validate their claims from different perspectives. This comprehensive approach strengthens the credibility of their findings.
3. The writing in this paper is clear and easy to understand, making it accessible to readers.

**Weaknesses:**

1. In the proposed five metrics, some formulas are not clearly defined. For instance, the meaning of \(\hat{ACC}\) in Eq. 1 and \(\bar{ACC}\) in Eq. 2 is not explicitly explained. Providing clearer definitions for these terms would enhance the understanding of the metrics and their implications in the context of the research.
2. In the tables, it would be helpful to use bold text or dashed lines to highlight specific data points, emphasizing the advantages of certain algorithms under particular settings.

**Questions:**

See Weaknesses

---

> ### Author Response · Authors · 2024-11-24
> **Official Response to Reviewer Mtt1**
>
> **Q1**: In the proposed five metrics, some formulas are not clearly defined. For instance, the meaning of $\hat{ACC}$ in Eq. 1 and $\bar{ACC}$ in Eq. 2 is not explicitly explained. Providing clearer definitions for these terms would enhance the understanding of the metrics and their implications in the context of the research.
>
> **A1**: Thank you for your insightful comment regarding the clarity of the definitions for $\hat{ACC}$ in Eq. 1 and $\bar{ACC}$ in Eq. 2.
>
> To clarify:
> 1. **$\hat{ACC}$** in Eq. 1 represents the empirical accuracy corresponding to $r$, which is the observed performance of the model on the test set for a given $r$ value.
> 2. **$\bar{ACC}$** in Eq. 2 represents the average accuracy of the model across all values of $r$, providing a summary of the model's overall performance. It serves as a reference point for comparing the deviations (in terms of $|\hat{ACC} - \bar{ACC}|$ at different values of $r$.
>
> We acknowledge that these terms were not explicitly explained in the original submission, and we have ensured they are clearly defined in the revised manuscript to enhance the clarity and accessibility of the work.
>
> We appreciate your constructive feedback and hope this explanation resolves the issue.
>
> **Q2**: In the tables, it would be helpful to use bold text or dashed lines to highlight specific data points, emphasizing the advantages of certain algorithms under particular settings.
>
> **A2**: Thank you for your valuable suggestion regarding the presentation of data in the tables.
>
> We agree that emphasizing specific data points using bold text or dashed lines can significantly enhance readability and make the advantages of certain algorithms under particular settings more apparent.
>
> Specifically, we have used bold text to highlight the best-performing results for each metric and setting in the revised manuscript.

---

> ### Author Response · Authors · 2024-11-26
> **A gentle reminder: please respond to our rebuttal**
>
> Dear Reviewer Mtt1,
>
> Thank you for your time and effort in reviewing our work. We have carefully considered your detailed comments and questions, and we have tried to address all your concerns accordingly.
>
> As the deadline for revising manuscript is approaching, could you please go over our responses? If you find our responses satisfactory, we hope you could consider adjusting your initial rating. Please feel free to share any additional comments you may have.
>
> Thank you!
>
> Authors

---

> > ### Comment · Reviewer_Mtt1 · 2024-11-26
> >
> > Thank you for your response, which has resolved my concerns. After considering the feedback from other reviewers, I decide to maintain my score.

---

> > > ### Author Response · Authors · 2024-11-27
> > > **Thanks for your recognition and recommendation for our work**
> > >
> > > Dear Reviewer Mtt1,
> > >
> > > Thank you for your recognition and recommendation for our work. We are pleased that our response has successfully addressed all your concerns. If there are remaining issues or questions regarding our paper, we would be more than happy to address them to further clarify our contributions.
> > >
> > > Best regards,
> > >
> > > Authors

---

### Author Response · Authors · 2024-11-24
**General Response**

We thank the reviewers for their thorough and constructive comments. We are glad that the reviewers agree that our motivation is clear and interesting (all the reviewers mentioned). Our motivation is new to the SSL field (Reviewer wdK8 mentioned). Reviewers also pointed out that our experimental evaluation is solid and comprehensive (Mtt1, 244R, wdK8), our experimental analysis is very comprehensive (wdK8), providing some guidance for understanding the working mechanism of existing methods and designing new ones (CS3Q), and our presentation is clear (Mtt1, wdK8).


Based on the reviewers' valuable feedback, we have conducted a number of additional experiments, which hopefully resolve the reviewers’ concerns. In this revised version, we have also updated the manuscript and Appendix, where we highlight modifications with blue color. The major additional experiments and improvements are as follows:

1. We expanded the number of datasets to include more datasets across various modalities, including image, text, and tabular data, to ensure a more comprehensive evaluation.
2. We provided a concise summary of each indicator with their range, interpretation, and significance for evaluating model performance.
3. We have provided clearer definitions for several terms to enhance the understanding of the metrics.
4. We have used bold text to highlight the best-performing results for each metric and setting.
5. We conducted experiments over 10 times to ensure reliability and robustness of the results.
6. We introduced statistical significance test (p-value) to rigorously validate the experimental findings, further enhancing the credibility of the analysis.
7. We have added the standard deviations and the updated results for more comprehensive evaluation of the experimental results.
8. To enhance reproducibility, we have provided an anonymous link to our code at [https://anonymous.4open.science/r/RE-SSL-F034/README.md] and also added more experimental details. Our code will be public once our paper is accepted.

---

### Meta-Review · Area_Chair_TTcb · 2024-12-19

**Metareview:**

This paper challenges the assumption that unseen classes in unlabeled data harm the performance of semi-supervised learning (SSL) models, arguing that prior assessments were flawed. The authors propose a re-evaluation of existing safe SSL models by fixing the proportion of seen classes and introduce five evaluation metrics to accurately assess the influence of unseen classes on SSL models, thereby addressing the shortcomings of current evaluation schemes. By controlling variables and adjusting only the proportion of unseen classes, the study demonstrates that unseen classes may not degrade SSL performance and can even enhance it under certain conditions.

The experiments conducted are extensive and thorough, providing a comprehensive analysis of various safe SSL methods, and the clear writing style makes the paper accessible to readers, enhancing the overall credibility of the findings. However, the reviewers also raised concerns about the credibility of the experiments, noting that conducting experiments only on CIFAR-10 and CIFAR-100 limits the scope and reduces the generalizability of the results.

Based on the overall review, the paper is recommended for acceptance.

**Additional Comments On Reviewer Discussion:**

During the rebuttal period, the reviewers' opinions remained unchanged.

---

### Decision · Program_Chairs · 2025-01-22

Accept (Poster)